# Temporal and thermal profiling of the *Toxoplasma* proteome implicates parasite Protein Phosphatase 1 in the regulation of Ca$^{2+}$-responsive pathways

Alice L Herneisen[1,2], Zhu-Hong Li[3], Alex W Chan[1,3], Silvia NJ Moreno[3], Sebastian Lourido[1,2]*

[1]Whitehead Institute for Biomedical Research, Cambridge, United States; [2]Biology Department, Massachusetts Institute of Technology, Cambridge, United States; [3]Center for Tropical and Emerging Global Diseases, University of Georgia, Athens, United States

**Abstract** Apicomplexan parasites cause persistent mortality and morbidity worldwide through diseases including malaria, toxoplasmosis, and cryptosporidiosis. Ca$^{2+}$ signaling pathways have been repurposed in these eukaryotic pathogens to regulate parasite-specific cellular processes governing the replicative and lytic phases of the infectious cycle, as well as the transition between them. Despite the presence of conserved Ca$^{2+}$-responsive proteins, little is known about how specific signaling elements interact to impact pathogenesis. We mapped the Ca$^{2+}$-responsive proteome of the model apicomplexan *Taxoplasma gondii* via time-resolved phosphoproteomics and thermal proteome profiling. The waves of phosphoregulation following PKG activation and stimulated Ca$^{2+}$ release corroborate known physiological changes but identify specific proteins operating in these pathways. Thermal profiling of parasite extracts identified many expected Ca$^{2+}$-responsive proteins, such as parasite Ca$^{2+}$-dependent protein kinases. Our approach also identified numerous Ca$^{2+}$-responsive proteins that are not predicted to bind Ca$^{2+}$, yet are critical components of the parasite signaling network. We characterized protein phosphatase 1 (PP1) as a Ca$^{2+}$-responsive enzyme that relocalized to the parasite apex upon Ca$^{2+}$ store release. Conditional depletion of PP1 revealed that the phosphatase regulates Ca$^{2+}$ uptake to promote parasite motility. PP1 may thus be partly responsible for Ca$^{2+}$-regulated serine/threonine phosphatase activity in apicomplexan parasites.

*For correspondence:
lourido@wi.mit.edu

## Editor's evaluation

Herneisen et al., provide a comprehensive and thorough exploration of Ca$^{2+}$ responsive changes in the Toxoplasma proteome and the resulting phosphorylation events during the transition from intracellular residing parasites to egress from the host cell. Furthermore, a novel temperature stability profiling method of all proteins responding to Ca$^{2+}$ concentration with a change in stability is a novel applicable tool that here is used to map Ca$^{2+}$-responsive proteins in the parasites. They provide a compelling analysis of the complex data and carefully validate their findings using genetics and cell biology. This work is of the highest quality in the field.

## Introduction

Apicomplexan parasites cause persistent mortality and morbidity worldwide through diseases including malaria, toxoplasmosis, and cryptosporidiosis (*Havelaar et al., 2015*). The phylum member

*Toxoplasma gondii* alone infects >2 billion people. As obligate intracellular pathogens, apicomplexans are exquisitely tuned to transduce environmental signals into programs of motility, replication, and quiescence responsible for parasite pathogenesis and spread (*Bisio and Soldati-Favre, 2019*). $Ca^{2+}$ signals and their downstream effectors are a part of the signaling cascade that impacts almost every cellular function (*Lourido and Moreno, 2015*). Signaling begins with release of $Ca^{2+}$ from intracellular stores or influx through plasma membrane channels, resulting in diverse downstream events central to parasite virulence, including secretion of adhesive proteins, motility, and invasion into and egress from host cells. Together, these cellular processes orchestrate a dramatic transition from the replicative to the kinetic phase of the life cycle that allows parasites to spread to new host cells. Signal-transducing components downstream of $Ca^{2+}$ release are largely unknown yet are likely essential for apicomplexan viability and virulence (*Lourido and Moreno, 2015*; *Nagamune and Sibley, 2006*).

$Ca^{2+}$ can change a protein's state by direct binding or indirect effects such as triggering post-translational modification (PTM), interaction with other proteins, or relocalization. Indirect effects are fundamental to the propagation and amplification of signals across the $Ca^{2+}$-regulated network, and in most organisms, they are largely mediated by three classes of $Ca^{2+}$-binding proteins: $Ca^{2+}$-regulated kinases, $Ca^{2+}$-regulated phosphatases, and calmodulin (CaM) and related proteins (*Villalobo et al., 2019*). Genomic searches for canonical $Ca^{2+}$-binding domains in apicomplexans have identified several individual proteins involved in transducing and effectuating $Ca^{2+}$ signals (*Farrell et al., 2012*; *Huet et al., 2018*; *McCoy et al., 2017*), like kinases, phosphatases, and transporters (*Hortua Triana et al., 2018*; *Lourido et al., 2012*; *Lourido et al., 2010*; *Luo et al., 2005*; *Márquez-Nogueras et al., 2021*). However, many of the key signaling elements involved in fundamental $Ca^{2+}$ responses—including the channels responsible for its stimulated release—are either missing from apicomplexan genomes or have diverged beyond recognition, suggesting that eukaryotic pathogens evolved novel pathways for $Ca^{2+}$ mobilization and transduction (*Billker et al., 2009*; *Lourido and Moreno, 2015*).

In apicomplexans, $Ca^{2+}$-dependent protein kinases (CDPKs) have garnered the most attention as the only known $Ca^{2+}$-regulated kinases in the phylum (*Billker et al., 2009*). These kinases possess intrinsic $Ca^{2+}$-binding sites and do not rely on CaM-like mammalian CaMKs. In *T. gondii*, several of these CDPKs trigger parasite motility (*Lourido et al., 2012*; *Lourido et al., 2010*; *McCoy et al., 2012*; *Smith et al., 2022*; *Treeck et al., 2014*). The roles of *Tg*CDPK7 in replication and *Tg*CDPK2 in amylopectin granule formation also implicate $Ca^{2+}$ signaling in cellular functions outside the kinetic phase of the parasite lytic cycle (*Morlon-Guyot et al., 2014*; *Bansal et al., 2021*; *Uboldi et al., 2015*). Dephosphorylation has garnered comparatively little attention in these parasites (*Yang and Arrizabalaga, 2017*). The roles of the prototypical CaM and the $Ca^{2+}$/CaM-dependent phosphatase calcineurin have only been phenotypically examined, and their client proteins remain largely unknown (*Paul et al., 2015*; *Philip and Waters, 2015*). Although key players are conserved and essential across the Apicomplexa, no systematic efforts have been undertaken to globally map the $Ca^{2+}$ signaling pathways of these pathogens.

$Ca^{2+}$ signaling pathways have been repurposed in apicomplexans to regulate parasite-specific cellular processes governing the transition between the replicative and kinetic phases of the infectious cycle (*Brown et al., 2020*; *Pace et al., 2020*). Despite the presence of conserved $Ca^{2+}$-responsive proteins (*Lourido and Moreno, 2015*; *Nagamune and Sibley, 2006*), uncovering the $Ca^{2+}$ signaling architecture of apicomplexans demands a reevaluation of the entire network to understand how specific signaling elements interact to impact pathogenesis. We present an atlas of $Ca^{2+}$-regulated proteins in the model apicomplexan *T. gondii*, assembled from high-dimensional proteomic datasets. The physiological changes associated with stimulated motility in the asexual stages of parasites have been characterized for decades (*Blader et al., 2015*). Our approach identified at once hundreds of molecular components underpinning these processes. We find numerous $Ca^{2+}$-responsive proteins that are not predicted to bind $Ca^{2+}$, yet operate at critical junctures in the parasite signaling network. From this analysis, the protein phosphatase PP1 emerges as an unanticipated $Ca^{2+}$-responsive protein.

## Results

### Sub-minute phosphoproteomics reveals the topology of Ca²⁺-dependent signaling processes

We can emulate the endogenous signaling pathways that mediate the kinetic phase of the lytic cycle by treating isolated parasites with zaprinast, which stimulates PKG activation by inhibiting phosphodiesterases that degrade cGMP (*Brown et al., 2016*; *Lourido et al., 2012*). Zaprinast-stimulated motility occurs rapidly and in defined sequence in apicomplexans: initially, an increase in cGMP activates parasite protein kinase G (PKG), which phosphorylates substrates and stimulates Ca²⁺ release from internal stores (*Brown et al., 2020*; *Lourido and Moreno, 2015*). PKG likely performs functions that extend beyond regulating Ca²⁺ stores, for example by mobilizing diacylglycerol and phosphatidic acid (*Lourido et al., 2012*; *Brown et al., 2017*; *Bullen et al., 2016*; *Bisio et al., 2019*); however, the use or phosphodiesterase inhibitors like zaprinast allows us to stimulate endogenous Ca²⁺ release without flooding the cell with Ca²⁺, as is the case with ionophores. Ca²⁺-dependent protein kinases, such as *Tg*CDPK1 and *Tg*CDPK3, synergize with PKG to effectuate microneme secretion and parasite motility (*Brown et al., 2016*; *Lourido et al., 2012*; *Lourido et al., 2010*; *McCoy et al., 2012*). This signaling cascade is active within seconds of cGMP elevation; however, existing *T. gondii* phosphoproteomes compare changes at a single time point following Ca²⁺-ionophore stimulation (*Treeck et al., 2014*), providing only a snapshot of diverging signaling states. Here, we add kinetic resolution to these signaling pathways. We quantified dynamic changes in the phosphoproteome within a minute of stimulation with zaprinast and thus activation of the cGMP/Ca²⁺ pathway.

We collected five timepoints in the 60 s following stimulation (0, 5, 10, 30, and 60 s), as well as three DMSO-treated matched timepoints (0, 10, and 30 s), in biological duplicate (*Figure 1A*). Using TMTpro labeling methods (*Li et al., 2020*), we multiplexed 16 samples, allowing us to analyze a complete time course, with replicates and controls, in a single MS experiment. Phosphopeptides were enriched from the rest of the sample using sequential metal-oxide affinity chromatography, which maximizes phosphopeptide capture (*Tsai et al., 2014*). Our experiments quantified 4,055 parasite proteins, none of which exhibited more than a twofold change in abundance in the 60 s following stimulation (*Figure 1—figure supplement 1*).

Given the paucity of known phosphoreglatory interactions in apicomplexans compared to other organisms (*Weiss et al., 2020*), we employed several analysis approaches to maximize the identification of changing phosphosites. We first calculated phosphoregulation scores by summing peptide abundances of vehicle (DMSO) and zaprinast-treated samples, taking their ratios, and standardizing the values with a modified Z score (*Figure 1B*). From a phosphoproteome of 11,755 unique peptides with quantification values (belonging to 2,690 phosphoproteins), 839 phosphopeptides increased in abundance three modified Z scores above the median, whereas 154 decreased 1.8 Z scores below the median. Principal component analysis on the significant peptides distinguished the agonist treatment and time-course kinetics in the two principal components accounting for the greatest variability in the data (*Figure 1C* and *Figure 1—figure supplement 1*).

### Kinetically resolved clusters reveal regulatory subnetworks during zaprinast stimulation

We leveraged the kinetic resolution of our comprehensive phosphoproteomics datasets to identify subregulatory networks. A Gaussian mixture-model clustering algorithm (*Invergo et al., 2017*) heuristically resolved four clusters for phosphopeptides arising from zaprinast treatment: three clusters increasing with different kinetics, and one decreasing (*Figure 1D*). On average, the 173 phosphopeptides belonging to cluster 1 increased sharply in abundance within 5 s of treatment and continued to increase for the remainder of the time course (*Figure 1D*), suggesting that they belonged to the first wave of phosphoregulation. This cluster was enriched for phosphoproteins associated with phosphodiesterase activity, phospholipid binding, and Ca²⁺ binding (*Figure 1E*), including PDE1, PDE2, PI4,5K, PI3,4K, PI-PLC, a phosphatidylinositol-3,4,5-triphosphate 5-phosphatase, a putative Sec14, *Tg*CDPK2A and *Tg*CDPK7, and PPM2B (*Table 1* and *Figure 1F*).

Peptides in clusters 2 and 3 (173 and 527, respectively) increased more gradually and exhibited lower fold-changes than cluster 1 sites (*Figure 1D*). Cluster 2 was notably enriched in proteins functioning in transport of monovalent ions and lipids, cyclase activity, and Ca²⁺ binding (*Figure 1E*). This

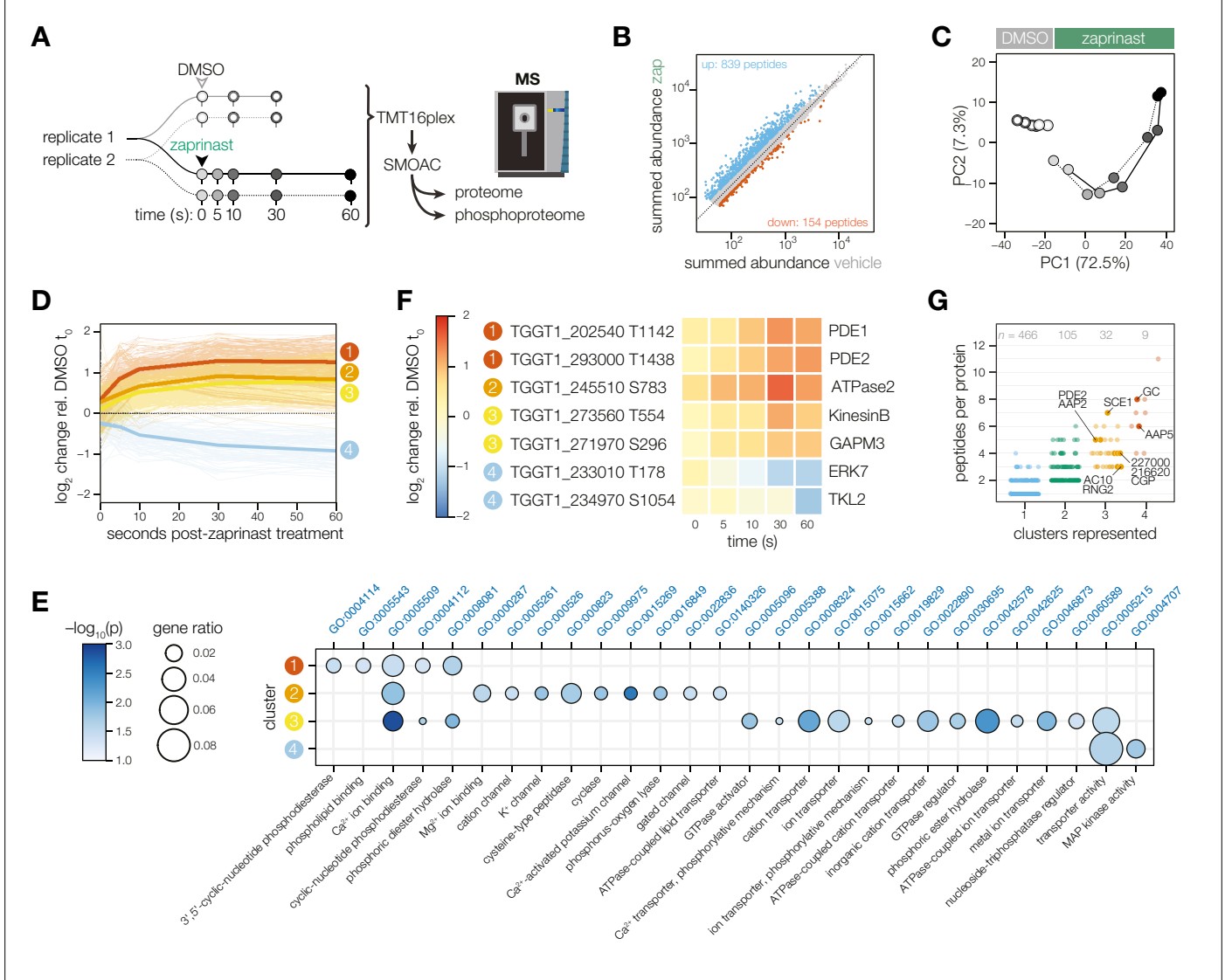

**Figure 1.** Phosphoregulation triggered by Ca²⁺ release. (**A**) Schematic of the sub-minute phosphoproteomics experiments with the Ca²⁺ signaling agonist zaprinast. (**B**) The summed abundances of unique phosphopeptides during zaprinast or vehicle (DMSO) treatment. The abundance ratios were transformed into a modified Z score and were used to threshold increasing (Z>3; *blue*) or decreasing (Z<−1.8; *orange*) phosphopeptides. (**C**) Principal component analysis of phosphopeptides identified as significantly changing. Symbols follow the schematic in A. (**D**) Gaussian mixture-model-based clustering of phosphopeptides changing during zaprinast treatment. Solid lines show the median relative abundance of each cluster. Opaque lines show the individual phosphopeptides belonging to each cluster. (**E**) GO terms enriched among phosphopeptides changing with zaprinast treatment, grouped by cluster. Gene ratio is the proportion of proteins with the indicated GO term divided by the total number of proteins belonging to each cluster. Significance was determined with a hypergeometric test; only GO terms with p<0.05 are shown. Redundant GO terms were removed. (**F**) Examples of phosphopeptides belonging to each cluster. (**G**) The number of clusters each phosphoprotein belongs to plotted against the number of changing phosphopeptides belonging to each protein. Gene names or IDs indicate proteins discussed in the text.

The online version of this article includes the following figure supplement(s) for figure 1:

**Figure supplement 1.** Metrics describing the zaprinast-dependent phosphoproteome.

set includes two putative Ca²⁺-activated K⁺ channels and the sodium-hydrogen exchangers NHE1 and NHE3 (*Table 1*). ATPase2 and the guanylyl cyclase in this cluster both have putative phospholipid-transporting ATPase domains. PI4,5K, PI3,4K, and *Tg*CDPK2A, noted in cluster 1, have additional phosphoregulatory sites belonging to cluster 2. Clusters 2 and 3 are enriched for phosphoproteins with metal transporter activity, including ATPases such as *Tg*A1, the putative copper transporter CuTP, a putative P5B-ATPase, a putative E1-E2 ATPase, and *Tg*ATP4. As observed in other *T. gondii*

**Table 1.** Gene IDs of proteins discussed in the text.

NS, not significant (using thresholds defined in the text). ND, not detected. TR, temperature range. CR, concentration range.

| Gene ID | Description in text | Reference | Phospho cluster | Thermal profiling |
|---|---|---|---|---|
| TGGT1_202540 | PDE1 | *Jia et al., 2017* | 1, 3 | NS |
| TGGT1_293000 | PDE2 | *Jia et al., 2017*; *Moss et al., 2022*; *Vo et al., 2020* | 1, 2, 4 | NS |
| TGGT1_245730 | PI4,5K | *Garcia et al., 2017* | 1, 2 | NS |
| TGGT1_276170 | PI3,4K | *Garcia et al., 2017* | 1, 2 | NS |
| TGGT1_248830 | PI-PLC | *Bullen et al., 2016*; *Fang et al., 2006* | 1, 3 | NS |
| TGGT1_288800 | Phosphatidylinositol-3,4,5-triphosphate 5-phosphatase | | 1 | NS |
| TGGT1_254390 | Putative Sec14 | | 1 | NS |
| TGGT1_206590 | CDPK2A | *Billker et al., 2009* | 1, 2, 3 | CR |
| TGGT1_228750 | CDPK7 | *Bansal et al., 2021* | 1, 3 | TR |
| TGGT1_267100 | PPM2B | *Yang et al., 2019*; *Yang and Arrizabalaga, 2017* | 1 | NS |
| TGGT1_238995 | $Ca^{2+}$-activated $K^+$ channel | | 2 | ND |
| TGGT1_273380 | $Ca^{2+}$-activated $K^+$ channel | | 2, 3 | ND |
| TGGT1_259200B | $Na^+/H^+$ exchanger | *Arrizabalaga et al., 2004* | 2, 3 | NS |
| TGGT1_305180 | $Na^+/H^+$ exchanger | *Francia et al., 2011* | 2, 3, 4 | NS |
| TGGT1_245510 | ATPase2 | *Chen et al., 2021* | 2 | ND |
| TGGT1_254370 | Guanylyl cyclase | *Bisio et al., 2019*; *Brown and Sibley, 2018* | 1, 2, 3, 4 | NS |
| TGGT1_312100 | Calcium ATPase TgA1 | *Luo et al., 2005*; *Luo et al., 2001* | 3 | ND |
| TGGT1_201150 | Copper transporter CuTP | *Kenthirapalan et al., 2014* | 3 | NS |
| TGGT1_318460 | Putative P5B-ATPase | *Møller et al., 2008* | 2 | NS |
| TGGT1_289070 | Putative E1-E2 ATPase | | 3 | NS |
| TGGT1_278660 | TgATP4 | *Lehane et al., 2019* | 3, 4 | NS |
| TGGT1_226020 | MFS transporter | | 3 | NS |
| TGGT1_230570 | MFS transporter | | 3 | NS |
| TGGT1_253700 | MFS transporter | | 3, 4 | NS |
| TGGT1_257530 | Tyrosine transporter ApiAT5-3 | *Parker et al., 2019*; *Wallbank et al., 2019* | 3, 4 | NS |
| TGGT1_292110 | Formate transporter TgFNT2 | *Erler et al., 2018*; *Zeng et al., 2021* | 3 | NS |
| TGGT1_270865 | Adenylyl cyclase | *Brown and Sibley, 2018*; *Jia et al., 2017* | 2, 3 | NS |
| TGGT1_238390 | Unique guanylyl cyclase organizer UGO | *Bisio et al., 2019* | 2, 3 | NS |
| TGGT1_309190 | ARO-interacting protein (adenylyl cyclase organizer) AIP | *Mueller et al., 2016*; *Mueller et al., 2013* | 1, 3 | NS |
| TGGT1_273560 | Divergent kinesin KinesinB | *Leung et al., 2017* | 2, 3 | NS |
| TGGT1_201230 | Divergent kinesin | *Wickstead et al., 2010* | 3 | ND |
| TGGT1_247600 | Dynein light chain | | 3 | NS |
| TGGT1_255190 | MyoC | *Frénal et al., 2017* | 3 | NS |
| TGGT1_278870 | MyoF | *Heaslip et al., 2016*; *Jacot et al., 2013* | 1, 2, 3 | NS |
| TGGT1_257470 | MyoJ | *Frénal et al., 2014* | 3 | CR |
| TGGT1_213325 | Uncharacterized TBC domain protein | | 3 | ND |

*Table 1 continued on next page*

*Table 1 continued*

| Gene ID | Description in text | Reference | Phospho cluster | Thermal profiling |
|---|---|---|---|---|
| TGGT1_221710 | Uncharacterized TBC domain protein | | 3 | NS |
| TGGT1_237280 | Uncharacterized TBC domain protein | | 1, 3, 4 | NS |
| TGGT1_274130 | Uncharacterized TBC domain protein | | 2, 3, 4 | NS |
| TGGT1_289820 | Uncharacterized TBC domain protein | | 3 | NS |
| TGGT1_206690 | GAPM2B | *Harding et al., 2019* | 3 | NS |
| TGGT1_271970 | GAPM3 | *Harding et al., 2019* | 3 | NS |
| TGGT1_233010 | ERK7 | *O'Shaughnessy et al., 2020* | 3, 4 | CR |
| TGGT1_234970 | Tyrosine kinase-like protein TgTLK2 | *Varberg et al., 2018* | 4 | NS |
| TGGT1_202900 | Putative K⁺ voltage-gated channel complex subunit | | 4 | NS |
| TGGT1_228200 | Vacuolar (H⁺)-ATPase G subunit | | 1, 4 | NS |
| TGGT1_233130 | Putative nucleoside transporter | | 4 | NS |
| TGGT1_258700 | MFS family transporter | | 4 | ND |
| TGGT1_269260 | SCE1 | *McCoy et al., 2017* | 1, 2, 3 | NS |
| TGGT1_295850 | AAP2 | *Engelberg et al., 2020* | 1, 2, 4 | CR |
| TGGT1_319900 | AAP5 | *Engelberg et al., 2020* | 1, 2, 3, 4 | NS |
| TGGT1_227000 | Apical polar ring protein | *Koreny et al., 2021* | 1, 3, 4 | ND |
| TGGT1_244470 | RNG2 | *Katris et al., 2014* | 1, 2, 3 | ND |
| TGGT1_292950 | Apical cap protein AC10 | *Back et al., 2020*; *Tosetti et al., 2020* | 1, 2, 3 | NS |
| TGGT1_240380 | Conoid gliding protein CGP | *Li et al., 2022* | 1, 3, 4 | NS |
| TGGT1_216620 | Ca²⁺ influx channel with EF hands | *Chang et al., 2019* | 1, 3, 4 | NS |
| TGGT1_246930 | Calmodulin-like protein CAM1 | *Long et al., 2017b* | ND | TR |
| TGGT1_262010 | ACalmodulin-like protein CAM2 | *Long et al., 2017b* | ND | TR, CR |
| TGGT1_216080 | apical lysine methyltransferase (AKMT) | *Heaslip et al., 2011* | NS | TR |
| TGGT1_270690 | DrpC | *Heredero-Bermejo et al., 2019*; *Melatti et al., 2019* | NS | TR |
| TGGT1_201880 | TgQCR9 | | ND | TR |
| TGGT1_204400 | ATP synthase subunit alpha | *Hayward et al., 2021*; *Huet et al., 2018*; *Mühleip et al., 2021*; *Salunke et al., 2018*; *Seidi et al., 2018* | NS | TR |
| TGGT1_231910 | ATP synthase subunit gamma | *Huet et al., 2018*; *Mühleip et al., 2021*; *Salunke et al., 2018*; *Seidi et al., 2018* | ND | TR |
| TGGT1_208440 | ATP synthase subunit 8/ASAP-15 | *Huet et al., 2018*; *Mühleip et al., 2021*; *Salunke et al., 2018*; *Seidi et al., 2018* | ND | TR, CR |
| TGGT1_215610 | ATP synthase subunit f/ICAP11/ASAP-10 | *Huet et al., 2018*; *Mühleip et al., 2021*; *Salunke et al., 2018*; *Seidi et al., 2018* | ND | TR |
| TGGT1_263080 | ATP synthase-associated protein ASAP-18/ATPTG14 | *Huet et al., 2018*; *Mühleip et al., 2021*; *Salunke et al., 2018*; *Seidi et al., 2018* | ND | TR, CR |
| TGGT1_246540 | ATP synthase-associated protein ATPTG1 | *Mühleip et al., 2021* | NS | TR |
| TGGT1_249240 | CaM | *Paul et al., 2015* | ND | CR |
| TGGT1_213800 | CnB | *Paul et al., 2015* | NS | CR |
| TGGT1_227800 | Eps15 | *Birnbaum et al., 2020*; *Chern et al., 2021* | NS | CR |
| TGGT1_269442 | ELC1 | *Nebl et al., 2011* | ND | CR |

*Table 1 continued on next page*

*Table 1 continued*

| Gene ID | Description in text | Reference | Phospho cluster | Thermal profiling |
|---|---|---|---|---|
| TGGT1_297470 | MLC1 | *Gaskins et al., 2004* | ND | CR |
| TGGT1_297470 | MLC5 | *Graindorge et al., 2016* | ND | CR |
| TGGT1_315780 | MLC7 | *Graindorge et al., 2016* | ND | CR |
| TGGT1_226030 | PKA-C1 | *Jia et al., 2017*; *Uboldi et al., 2018* | NS | CR |
| TGGT1_242070 | PKA-R | *Jia et al., 2017*; *Uboldi et al., 2018* | NS | CR |
| TGGT1_210830 | Putative RIO1 kinase | | NS | CR |
| TGGT1_310700 | PP1 | *Paul et al., 2020*; *Zeeshan et al., 2021* | ND | CR |
| TGGT1_207910 | Calcium-hydrogen exchanger TgCAX | *Guttery et al., 2013* | NS | CR |
| TGGT1_311080 | Apicoplast two-pore channel TgTPC | *Li et al., 2021* | NS | CR |
| TGGT1_204050 | Subtilisin 1 SUB1 | | NS | CR |
| TGGT1_206490 | Metacaspase 1 with a C2 domain | *Li et al., 2015* | ND | CR |
| TGGT1_310810 | $Ca^{2+}$-activated apyrase | | ND | CR |
| TGGT1_321650 | RON13 | *Lentini et al., 2021* | ND | CR |
| TGGT1_286710 | Uncharacterized metal-binding protein with zinc fingers | | ND | CR |
| TGGT1_309290 | Uncharacterized metal-binding protein with HD domain | | ND | CR |
| TGGT1_225690 | Apical cap protein AC7 | *Chen et al., 2015* | 2, 3 | NS |
| TGGT1_234250 | CHP interacting protein CIP1 | *Long et al., 2017a* | NS | ND |

$Ca^{2+}$-stimulated phosphoproteomes, at later time points small-molecule transporters are phosphorylated, including MFS transporters, ApiAT5-3, and *Tg*FNT2. The guanylyl and adenylyl cyclases are also extensively modified, as are the cyclase organizers UGO and AIP.

Cluster 3, which represents a latter wave of phosphoregulation, is uniquely enriched in proteins functioning in subcellular remodeling, vesicle trafficking, and glideosome activity (*Figure 1E*). This set includes two divergent kinesins (KinesinB and TGGT1_201230), a dynein light chain, and the myosin motors MyoC, MyoF, and MyoJ (*Table 1*). Several uncharacterized GTPase regulators are phosphorylated later in the zaprinast response, including putative ARF1 activators (TGGT1_225310 and TGGT1_266830) as well as five uncharacterized TBC domain proteins. The glideosome-associated membrane proteins GAPM2B and GAPM3, which link the IMC, alveolin network, and microtubules, are also dynamically phosphorylated.

Cluster 4, the only cluster characterized by decreasing phosphorylation, was functionally enriched in phosphoproteins involved in transporter and MAP kinase activity. ERK7 was dephosphorylated within 30 s of zaprinast treatment, whereas *Tg*TLK2 was dephosphorylated only at the final 60 s time point. ERK7 regulates conoid and cytoskeletal stability during cell division, with secondary functions in parasite egress, motility, and invasion (*Back et al., 2020*; *Dos Santos Pacheco et al., 2021*; *O'Shaughnessy et al., 2020*). TLK2 regulates parasite replication (*Smith et al., 2022*; *Varberg et al., 2018*). Several phosphoproteins functioning in ion transport belonged to cluster 4, including a putative $K^+$ voltage-gated channel complex subunit, a putative Vacuolar (H+)-ATPase G subunit, *Tg*ATP4, and the guanylyl cyclase (*Table 1*), which has a P-type ATPase domain with unknown ion specificity. Small-molecule transporters included a putative nucleoside transporter, ApiAT5-3, and a MFS family transporter. Cluster 4 was the only class of peptides that was not functionally enriched in $Ca^{2+}$ binding proteins (*Figure 1E*).

We identified phosphoproteins with peptides belonging to several clusters (*Figure 1G*). Such proteins may have multiple phosphosites regulated with different kinetics by the same enzyme; or by

different enzymes that alight upon the target at varying spatiotemporal scales. For example, SCE1 and TGGT1_309910 have phosphopeptides belonging to all three increasing clusters, likely resulting from phosphorylation by *Tg*CDPK3 and PKG, respectively. Indeed, SCE1 was implied to be a *Tg*CDPK3 target through a genetic suppressor screen (*McCoy et al., 2017*). TGGT1_309910 is the ortholog of *Plasmodium falciparum Pf* ICM1, a PKG substrate identified through proteomic interaction studies (*Balestra et al., 2021*), although it remains uncharacterized in *T. gondii*. By contrast, several proteins belong to both increasing and decreasing clusters, likely targets of both kinases and phosphatases. We consider these proteins candidate signaling platforms. Several such proteins localize to discrete domains of the apical complex, including the guanylyl cyclase; apical annuli proteins AAP2 and AAP5; apical polar ring proteins TGGT1_227000 and RNG2; and the apical cap protein AC10 and a recently identified conoid gliding protein CGP (*Table 1*). Along with the guanylyl cyclase, the cAMP-specific phosphodiesterase PDE2 and a $Ca^{2+}$ influx channel with EF hands exhibit peptides belonging to both increasing and decreasing clusters. Our phosphoproteome thus identifies candidate mediators of the feedback between cyclic-nucleotide and $Ca^{2+}$ signaling.

The waves of phosphoregulation largely corroborate the sequence of physiological events observed following zaprinast treatment. The first cluster likely includes the most proximal targets of PKG. Previous PKG-dependent phosphoproteomes from the related apicomplexan *P. falciparum* similarly implicate PI-PLC as a substrate of the kinase (*Alam et al., 2015*; *Balestra et al., 2021*; *Brochet et al., 2014*). The product of PI-PLC activity, $IP_3$, stimulates $Ca^{2+}$ store release in parasites (*Garcia et al., 2017*). In turn, phosphoproteins in clusters 2, 3 and 4 likely contain the targets of $Ca^{2+}$-regulated kinases and phosphatases.

## Thermal profiling identifies $Ca^{2+}$-dependent shifts in *T. gondii* protein stability

The phosphoproteome identified numerous dynamic changes in response to $Ca^{2+}$ release. However, this information alone is not sufficient to infer the enzymes responsible for phosphorylation, or which events may be functionally relevant. Thermal proteome profiling (TPP) has been used to identify small molecule–target interactions in living cells and cell extracts (*Dai et al., 2019*; *Mateus et al., 2020*; *Savitski et al., 2014*). TPP operates on the premise that ligand binding induces a thermal stability shift, stabilizing, or destabilizing proteins that change conformationally in response to the ligand, and such changes in stability can be quantified by MS (*Dziekan et al., 2020*; *Reinhard et al., 2015*). Cells are incubated with different concentrations of ligand and heated, causing thermal denaturation of proteins. The soluble protein is extracted and quantified with multiplexed, quantitative methods, giving rise to thousands of thermal denaturation profiles. Proteins engaging the ligand are identified by their concentration-dependent thermal shift. We previously used this method to identify the target of the antiparasitic compound ENH1 (*Herneisen et al., 2020*; *Herneisen and Lourido, 2021*) and to measure changes to the *T. gondii* proteome when depleting the mitochondrial protein DegP2 (*Harding et al., 2020*). In a conceptual leap, we reasoned that TPP could also detect $Ca^{2+}$-responsive proteins if parasite extracts were exposed to different concentrations of $Ca^{2+}$, allowing us to systematically identify the protein components of signaling pathways on the basis of biochemical interactions with $Ca^{2+}$ and its effectors.

Intracellular free $Ca^{2+}$ levels span three orders of magnitude, from low nanomolar in the cytoplasm to high micromolar in organelles like the ER (*Lourido and Moreno, 2015*). To measure protein thermal stability at precisely defined $[Ca^{2+}]_{free}$, we combined crude parasite extracts with calibrated $Ca^{2+}$ buffers in a solution mimicking the ionic composition of the cytoplasm. As a proof of principle, we measured the thermal stability of the $Ca^{2+}$-dependent protein kinase *Tg*CDPK1, as the conformational changes of this enzyme have been structurally characterized (*Ingram et al., 2015*; *Wernimont et al., 2010*). Parasite lysates were adjusted to 10 different concentrations of free $Ca^{2+}$ and heated to 58 °C, based on prior estimates of the melting temperature of *Tg*CDPK1 (*Herneisen et al., 2020*). As measured by immunoblot band intensity, *Tg*CDPK1 was strongly stabilized by $Ca^{2+}$ (*Figure 2A*), suggesting that our experimental system is sensitive to $Ca^{2+}$-dependent stability changes. The calculated $EC_{50}$ using this approach was in the low μM range, consistent with studies using recombinant enzymes (*Ingram et al., 2015*; *Wernimont et al., 2010*).

The effect of $Ca^{2+}$ on the global thermostability of the proteome has not been assessed. Therefore, we first generated thermal profiles of the *T. gondii* proteome without or with 10 μM $Ca^{2+}$, which

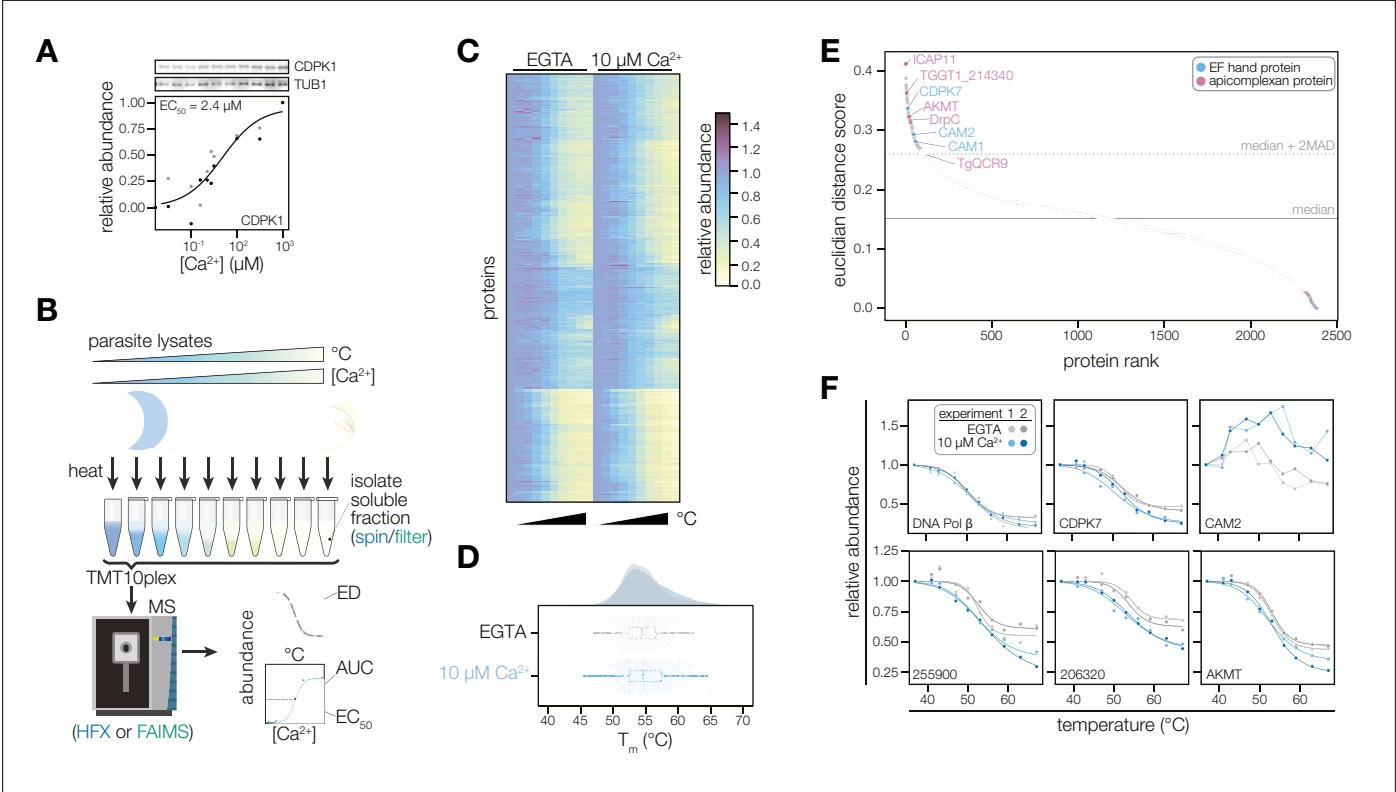

**Figure 2.** Thermal profiling identifies proteins that change stability in response to $Ca^{2+}$. (**A**) Thermal shift assays can detect $Ca^{2+}$-dependent stability of CDPK1 in extracts. Parasite lysates were combined with 10 concentrations of $Ca^{2+}$ spanning the nanomolar to micromolar range. After denaturation at 58 °C, the soluble fraction was separated by SDS-PAGE and probed for CDPK1. Band intensity was normalized to the no-$Ca^{2+}$ control and scaled. Points in shades of gray represent two different replicates. A dose-response curve was calculated for the mean abundances. (**B**) Schematic of the thermal profiling workflow. In the temperature-range experiment, parasite lysates were combined with EGTA or 10 µM $[Ca^{2+}]_{free}$ and heated at 10 temperatures spanning 37–67°C. In the concentration-range experiment, parasite lysates were combined with 10 different $[Ca^{2+}]_{free}$ (nM–mM range) and heated at 50, 54, or 58 °C. Temperature-range shifts were quantified by the Euclidean distance (ED) score, a weighted ratio of thermal stability differences between treatments and replicates. Concentration-range shifts were summarized by pEC$_{50}$, area under the curve (AUC), and goodness of fit ($R^2$). (**C**) Heat map of protein thermal stability relative to the lowest temperature (37 °C) in 0 or 10 µM $Ca^{2+}$. The mean relative abundance at each temperature was calculated for 2381 proteins. Proteins are plotted in the same order in both treatments. (**D**) Raincloud plots summarizing the distribution of $T_m$ in lysates with EGTA (gray) or 10 µM $[Ca^{2+}]_{free}$ (blue). The average melting temperatures of proteins identified in two replicates were plotted. (**E**) Proteins rank-ordered by euclidean distance score quantifying the $Ca^{2+}$-dependent shift in thermal stability. Solid and dotted lines represent the median ED score and two modified Z scores above the median, respectively. Highlighted proteins have EF hand domains (blue) or are conserved in apicomplexans (pink). (**F**) Thermal profiles of individual proteins: DNA polymerase β (TGGT1_233820); the EF hand domain-containing proteins CDPK7 (TGGT1_228750) and the calmodulin-like protein CAM2 (TGGT1_262010); potential $Ca^{2+}$-leak channels TGGT1_255900 and TGGT1_206320; and AKMT (TGGT1_216080).

The online version of this article includes the following source data and figure supplement(s) for figure 2:

**Source data 1.** This file contains the source data that was quantified to make the graph presented in *Figure 2*.

**Source data 2.** This file contains the source data that was quantified to make the graph presented in *Figure 2*.

**Source data 3.** This file contains the annotated source data that was quantified to make the graph presented in *Figure 2*.

**Figure supplement 1.** Extended data for thermal profiling experiments.

is representative of the resting and stimulated $Ca^{2+}$ concentrations of cell cytoplasm (*Lourido and Moreno, 2015*). A thermal challenge between 37 and 67°C induced denatured aggregates, which were separated from stable proteins by ultracentrifugation. The soluble fraction was digested and labeled with isobaric mass tags, pooled, fractionated, and analyzed with an orbitrap mass spectrometer (*Figure 2B*). We detected 3,754 proteins, of which 2,381 yielded thermal profiles for both conditions in both replicates (*Figure 2C* and *Figure 2—figure supplement 1*); the remaining proteins were not detected in all experiments. The median melting temperatures—the temperatures at which proteins are 50% denatured—were 54.4 and 54.7°C in the lysates without and with $Ca^{2+}$, respectively. The distribution of melting temperatures was largely overlapping in the two conditions (*Figure 2D*),

suggesting that $Ca^{2+}$-dependent changes in protein stability were restricted to a subset of proteins. We additionally calculated an area under the curve (AUC) metric by numerical integration using the trapezoidal rule (*Herneisen and Lourido, 2021*) to compare the stabilities of proteins with atypical melting behavior (*Figure 2—figure supplement 1*), such as components of the tubulin cytoskeleton or parasite conoid.

To discover proteins with $Ca^{2+}$-dependent stability shifts in the initial temperature range experiment, we rank-ordered proteins by Euclidean distance (ED) scores (*Dziekan et al., 2020*) quantifying the shift in thermal profiles with and without $Ca^{2+}$ (*Figure 2B and E*). The majority of proteins, such as DNA polymerase β (TGGT1_233820), exhibited similar melting behavior in both conditions (*Figure 2F*). Our analysis identified as $Ca^{2+}$-responsive parasite-specific proteins with EF hands, including *Tg*CDPK7 and the calmodulin-like proteins CAM1 and CAM2 (*Table 1*, *Figure 2F* and *Figure 2—figure supplement 1*). The ED metric identified the stability changes of both CAM proteins despite a lack of typical melting behavior, supporting the use of this statistic. Membrane proteins, including potential $Ca^{2+}$-leak channels localizing to the ER (*Barylyuk et al., 2020*; TGGT1_255900 and 206320), also exhibited thermal shifts (*Figure 2F*). Several proteins specific to the apicomplexan parasite phylum exhibited $Ca^{2+}$-regulation (*Barylyuk et al., 2020*; *Sidik et al., 2016a*), including an apical lysine methyltransferase (AKMT) that relocalizes during $Ca^{2+}$-stimulated egress *Heaslip et al., 2011*; DrpC, which regulates the stability of parasite organelles; a hypothetical protein (TGGT1_214340) with no annotated domains (*Figure 2—figure supplement 1*); and enzyme subunits involved in cellular metabolism.

The data also inform hypotheses about $Ca^{2+}$ homeostasis and energetics in the parasite mitochondrion and apicoplast. A divergent subunit of the mitochondrial complex III, *Tg*QCR9, as well as components of the ATP synthase complex, were destabilized by $Ca^{2+}$ (*Figure 2—figure supplement 1*). These include the ATP synthase subunits α, γ, 8/ASAP-15, and f/ICAP11/ASAP-10; and the ATP synthase-associated proteins ASAP-18/ATPTG14 and ATPTG1 (*Table 1*). TGGT1_209950, a conserved alveolate thioredoxin-like protein suggested to localize to the apicoplast by spatial proteomics (*Barylyuk et al., 2020*), was similarly destabilized by $Ca^{2+}$ (*Figure 2—figure supplement 1*). The apicomplexan mitochondrion has been reported to sense cytosolic $Ca^{2+}$ fluctuations (*Gazarini and Garcia, 2004*), and the apicoplast may take up $Ca^{2+}$ through direct contact sites with the ER (*Li et al., 2021*). Furthermore, redox signals were recently reported to induce parasite $Ca^{2+}$ signaling and motility (*Alves et al., 2021*; *Stommel et al., 1997*). Our resource identifies several proteins that may couple ion homeostasis and cellular metabolism. The recent structure of the apicomplexan ATP synthase suggests that all of the $Ca^{2+}$-responsive elements identified here, including subunits restricted to apicomplexan parasites, protrude into the mitochondrial matrix (*Mühleip et al., 2021*), which has elevated $Ca^{2+}$ (*Giorgi et al., 2018*). In the case of TGGT1_209950, plants and algae have $Ca^{2+}$-sensing thioredoxins (calredoxins) in chloroplasts (*Hochmal et al., 2016*) with $Ca^{2+}$-regulated activity mediated by EF hands (*Charoenwattanasatien et al., 2020*). TGGT1_209950 lacks EF-hands; however, structural modeling based on primary sequence and predicted secondary features (*Kelley et al., 2015*; *Meier and Söding, 2015*) suggested similarity to calsequestrin, which lacks a structured $Ca^{2+}$-binding motif. Our approach has thus identified candidate $Ca^{2+}$-responsive proteins in metabolic organelles that would have been missed by bioinformatic searches; although, additional experiments are needed to determine the interactions each of these candidates has with $Ca^{2+}$.

## Determining the specificity and sensitivity of $Ca^{2+}$-responsive proteins

The temperature range experiment has the advantage of generating complete thermal stability profiles, but does not inform the relative affinities of the $Ca^{2+}$-dependent stability change. To address this gap, we examined protein stability across 10 $Ca^{2+}$ concentrations (EGTA and 10 nM to 1 mM). Based on the temperature-range experiments, we selected thermal challenge temperatures of 50, 54, and 58 °C to target protein with a range of thermal stabilities in these dose-response experiments (*Figure 2—figure supplement 1*). We hypothesized that $Ca^{2+}$-responsive proteins would exhibit sigmoidal dose-response trends in thermal stability, similarly to *Tg*CDPK1 (*Figure 2A*). We performed four independent concentration-range experiments on two mass spectrometers using different separation methods (ultracentrifugation or filtration) to capture different types of aggregates.

The concentration range thermal-profiling experiments provide insight into the magnitude of the $Ca^{2+}$-dependent change (AUC parameter) and its dose-dependency ($EC_{50}$) (*Figure 2B*). Clear responses were observed for several known $Ca^{2+}$-binding enzymes with EF hands, including calmodulin,

calcineurin, and several parasite $Ca^{2+}$-dependent protein kinases (CDPKs; *Figure 3A*). These parameters can be used to classify proteins as stabilized (AUC >1; e.g., *Tg*CDPK1, *Tg*CDPK2A, and *Tg*CDPK3) or destabilized (AUC <1; e.g., calmodulin and calcineurin B) by $Ca^{2+}$ (*Figure 3A*). Furthermore, $EC_{50}$ measurements may inform specific hypotheses about a protein's involvement in signaling or homeostasis, based on the $Ca^{2+}$ concentration at which the protein is predicted to change.

Our results generated a de novo catalog of proteins with $Ca^{2+}$-dependent stability. Of the 4,403 proteins identified with sufficient quantification values for curve fitting, 228 were classified as $Ca^{2+}$-responsive based on exhibiting a dose-response trend ($R^2$) greater than 0.8 with a stability change (AUC) of two modified Z scores from the median (*Figure 3B*, *Figure 3—figure supplement 1*, and *Supplementary file 2*). Functional enrichment showed that such $Ca^{2+}$-responsive proteins were significantly enriched in $Ca^{2+}$-binding functions, protein kinase activity, and metal affinity (*Figure 3C*). We examined the 40 EF hand domain–containing proteins detected in our mass spectrometry experiments (*Figure 3D*), 25 of which exhibited dose-responsive behavior. Signaling and trafficking proteins were tuned to respond at lower $Ca^{2+}$ concentrations: *Tg*CDPK2A and CaM responded robustly with nM $EC_{50}$; *Tg*CDPK1, *Tg*CDPK3, CnB, and Eps15 exhibited low µM $EC_{50}$. In the cases of *Tg*CDPK1, CaM, and CnB, these values are consistent with the $Ca^{2+}$ activation constants determined from studies with recombinant proteins (*Feng and Stemmer, 2001*; *Feng and Stemmer, 1999*; *Wernimont et al., 2010*). By contrast, myosin-associated subunits responded at higher levels: Myosin light chains (ELC1, MLC1, MLC5, and MLC7) and the calmodulin-like proteins CAM1 and CAM2 responded above 10 µM $Ca^{2+}$ (*Table 1*). Two additional proteins (TGGT1_255660 and TGGT1_259710) consistently responded with µM $EC_{50}$ values but have not been functionally characterized (*Figure 3—figure supplement 1*). The remaining EF hand domain–containing proteins may bind $Ca^{2+}$ outside of the experimental concentration range; or may not undergo detectable stability changes upon $Ca^{2+}$ binding (*Mateus et al., 2020*).

Several signaling enzymes lacking canonical $Ca^{2+}$-binding domains exhibited a dose-response trend ($R^2$ >0.8) in our thermal profiling experiments (*Figure 3B and E*). The PKA regulatory and catalytic subunits (TGGT1_242070 and TGGT1_226030) responded robustly to $Ca^{2+}$ with 10–100 µM $EC_{50}$ values in several experiments. A putative RIO1 kinase (TGGT1_210830) also responded consistently to $Ca^{2+}$, although the magnitude of the change was small (*Figure 3—figure supplement 1*). The atypical kinase ERK7 was destabilized at high $Ca^{2+}$ concentrations (*Figure 3—figure supplement 1*), similarly to other $Ca^{2+}$-responsive proteins detected at the parasite apex. Rhoptry and dense granule kinases had $EC_{50}$ values in between 100 µM and 1 mM, consistent with the high concentration of $Ca^{2+}$ in the secretory pathway (*Figure 3E*). We also searched for $Ca^{2+}$-responsive phosphatases. The known $Ca^{2+}$-regulated phosphatase subunit, CnB, was destabilized (*Figure 3F*). Protein Phosphatase 1 (PP1) responded to $Ca^{2+}$ consistently (*Figure 3F* and 5A), although the catalytic subunit is not an intrinsic $Ca^{2+}$ sensor. Our resource thus places proteins without previously characterized $Ca^{2+}$ responses within the broader $Ca^{2+}$ signaling network.

Our search and annotation of $Ca^{2+}$-responsive proteins also identified candidates that link $Ca^{2+}$ and ion homeostasis. This list includes several transporters and channels (*Table 1*): *Tg*CAX, a protein with structural homology to LDL receptors (TGGT1_245610), two proteins with structural homology to the mitochondrial $Ca^{2+}$ uniporter (TGGT1_257040 and TGGT1_211710), and an apicoplast two-pore channel. Several $Ca^{2+}$-responsive hydrolases were also found, including SUB1, a metacaspase with a C2 domain, and a $Ca^{2+}$-activated apyrase. These proteins may function during the intracellular, replicative phase of the lytic cycle, for which the function of $Ca^{2+}$ is relatively unexplored.

Several divalent cations regulate protein structure and activity in addition to $Ca^{2+}$, including $Mg^{2+}$ and $Zn^{2+}$. Such metal-binding proteins might appear $Ca^{2+}$-responsive in our thermal profiling approach. We buffered our parasite extracts with excess $Mg^{2+}$ (1 mM) to mitigate non-specific changes caused by the concentration of divalent cations. Nevertheless, to determine whether our approach identified additional metal-binding proteins, we compared the $EC_{50}$ values of candidates that bind different divalent cations, based on the presence of Interpro domains and through manual annotation (*Figure 3—figure supplement 1*). The $EC_{50}$ values of putative $Mg^{2+}$-binding proteins were mostly in the mM range, which may result from nonspecific displacement of $Mg^{2+}$ by $Ca^{2+}$. However, a subset of $Mg^{2+}$-binding proteins were stabilized by micromolar $Ca^{2+}$–for instance, glucosephosphate-mutase (GPM1) family proteins TGGT1_285980 A and B. The GPM1 ortholog in *Paramecium*, parafusin (PFUS), is involved in $Ca^{2+}$-regulated exocytosis (*Subramanian and Satir, 1992*) A PFUS homolog in *T. gondii*

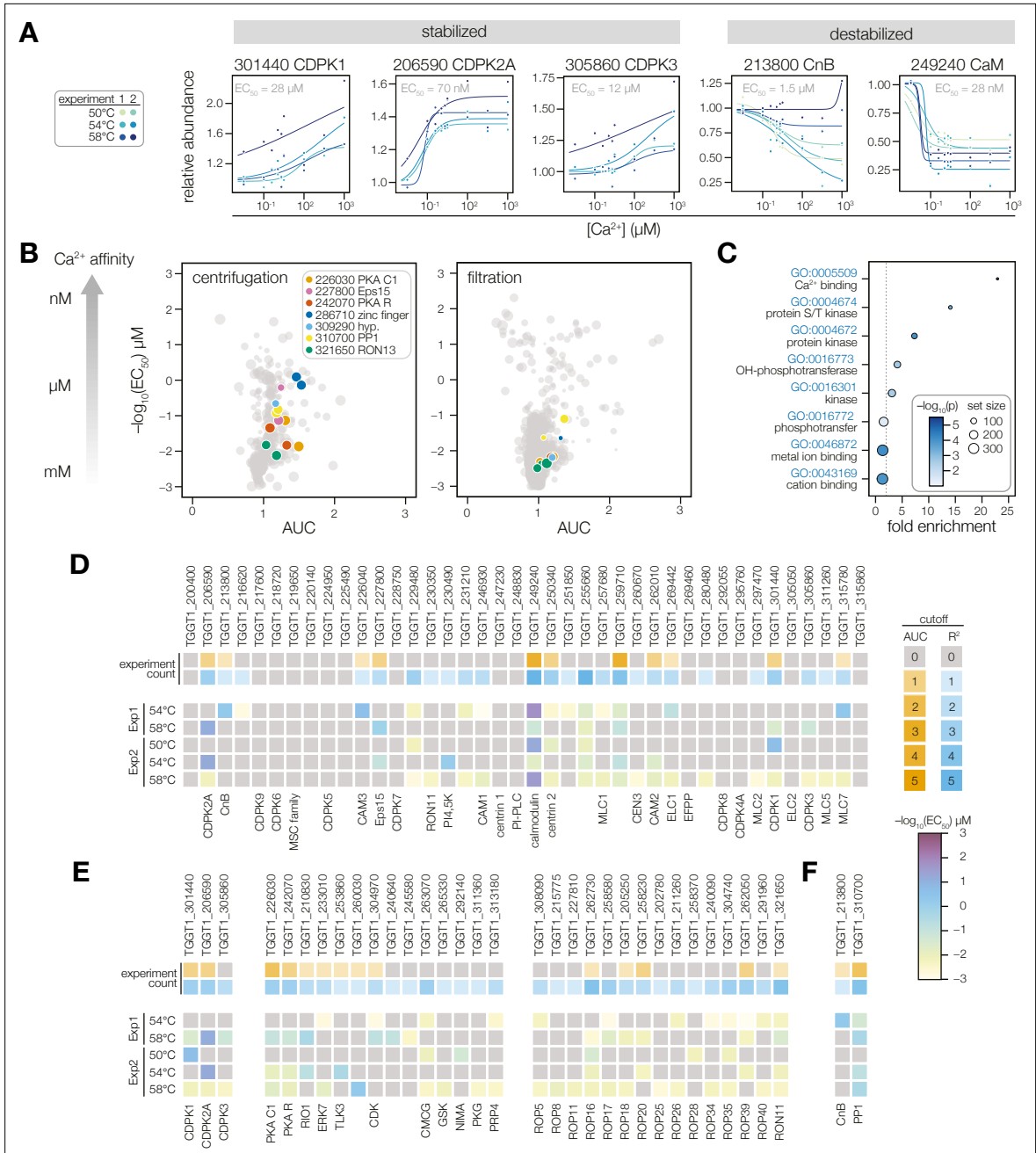

**Figure 3.** Thermal profiling identifies anticipated and unexplored $Ca^{2+}$-responsive proteins. (**A**) Mass spectrometry-derived thermal profiles of EF hand-containing proteins stabilized or destabilized by $Ca^{2+}$. Relative abundance is calculated relative to the protein abundance at 0 μM $Ca^{2+}$. $EC_{50}$ is the median of the $EC_{50}$ values of the curves displayed on the plots. (**B**) The magnitude of $Ca^{2+}$-dependent stabilization (AUC) plotted against the sensitivity ($pEC_{50}$) for protein abundances exhibiting a dose-response trend with an $R^2 > 0.8$. Point size is scaled to $R^2$. Summary parameters for the different separation methods (ultracentrifugation or filtration) are plotted separately. Colors identify candidates with $Ca^{2+}$-responsive behavior validated in *Figure 4*. (**C**) Gene ontology (GO) terms enriched among candidate $Ca^{2+}$-responsive proteins (AUC greater than two modified Z scores and $R^2$ dose-response >0.8). Fold enrichment is the frequency of $Ca^{2+}$-responsive proteins in the set relative to the frequency of the GO term in the population of detected proteins. Significance was determined with a hypergeometric test; only GO terms with p<0.05 are shown. (**D–F**) EF hand domain proteins (**D**), protein kinases (**E**), and protein phosphatases (**F**) detected in the thermal profiling mass spectrometry datasets. The top rows indicate if a protein passed the AUC cutoff (orange) or $R^2$ cutoff (blue) for dose-response behavior. The opacity of the band represents the number of experiments in which the protein exhibited the behavior (out of five). The five rows below summarize the $pEC_{50}$ of each experiment in which the protein exhibited a dose-response trend with $R^2 > 0.8$. Kinases are loosely grouped as CDPK's (included as a reference), non-rhoptry kinases, and secretory pathway kinases.

The online version of this article includes the following figure supplement(s) for figure 3:

**Figure supplement 1.** Extended analysis of thermal profiling experiments.

has been observed to co-localize with micronemes in the apical third of the cell (*Matthiesen et al., 2001*). Therefore, the thermal stabilization of GPM1 could arise from Ca$^{2+}$-dependent modifications to the enzyme (*Subramanian et al., 1994*; *Subramanian and Satir, 1992*).

## Validation of Ca$^{2+}$-dependent thermal stability

We selected five proteins exhibiting consistent Ca$^{2+}$-responsive behavior by MS for validation: an EF hand domain–containing protein (Eps15), two kinases (RON13 and PKA-C1), and two uncharacterized putative metal-binding proteins (TGGT1_286710 and TGGT1_309290). The first three candidates were selected for potential involvement in dynamic Ca$^{2+}$-regulated processes. Eps15 (TGGT1_227800) was recently shown to mediate endocytosis in *P. falciparum* (*Birnbaum et al., 2020*) and localized to puncta bridging the inner membrane complex (IMC) and cytoskeleton in *T. gondii* (*Chern et al., 2021*). PKA-C1 is thought to antagonize Ca$^{2+}$ signaling in *T. gondii* (*Jia et al., 2017*; *Uboldi et al., 2018*). RON13 is a secretory-pathway kinase that phosphorylates substrates in apicomplexan invasion-associated organelles called rhoptries (*Lentini et al., 2021*). The function of the remaining two proteins is unknown, although TGGT1_286710 contains a zinc finger domain and TGGT1_309290, annotated as a hypothetical protein, contains an HD/PDEase domain—both contribute to parasite fitness in cell culture (*Sidik et al., 2016a*).

We appended epitope tags to the endogenous locus of each candidate. These Ca$^{2+}$-responsive proteins localized to diverse structures (*Figure 4B*). PKA-C1 resided at the parasite pellicle, as previously reported (*Jia et al., 2017*; *Uboldi et al., 2018*). Eps15 concentrated at puncta in the apical third of the parasite. TGGT1_286710 and TGGT1_309290 localized to the parasite nucleus, with TGGT1_286710 exhibiting additional staining in the residual body. Finally, RON13 appeared to stain the rhoptries (*Lentini et al., 2021*). To validate the Ca$^{2+}$-dependent stability of individual candidates, we prepared parasite extracts as described earlier, but relied on an immunoblot readout instead of MS. In these five cases, stabilization of the candidates was confirmed in multiple biological replicates. Several controls (TUB1, MIC2, GAP45, and SAG1) were stable across all Ca$^{2+}$ concentrations tested (*Figure 4* and *Figure 4—figure supplement 1*). In the case of PKA-C1, Eps15, and both putative metal-binding proteins, the immunoblot experiments revealed an even higher EC$_{50}$ than was measured in the MS experiments. We conclude that the stability changes detected by our global proteomics methods are robust, although the precise features of Ca$^{2+}$-dependent stability may differ based on the method used to assess them.

## A PP1 holoenzyme performs Ca$^{2+}$-responsive functions required for parasite spread

Our orthogonal proteomics approaches map Ca$^{2+}$-responsive phosphorylation and thermal stability. Unexpectedly, the catalytic subunit of Protein Phosphatase 1 (PP1, TGGT1_310700) exhibited consistent stabilization by Ca$^{2+}$ in our mass spectrometry experiments (*Figure 3F* and *Figure 5A*). The contribution of phosphatases to Ca$^{2+}$ signaling in apicomplexans is poorly understood (*Yang and Arrizabalaga, 2017*) and the Ca$^{2+}$-responsive behavior of PP1 has not been reported in other eukaryotes. The function of the phosphatase has not been directly studied in *T. gondii*, although PP1 inhibitors have been shown to block invasion of host cells (*Delorme et al., 2002*), and recent experiments in *P. falciparum* suggest that PP1 is required for the merozoite egress-to-invasion transition (*Paul et al., 2020*). Intriguingly, PP1 was recently observed to relocalize to the apical complex in highly motile *Plasmodium* berghei ookinetes (*Zeeshan et al., 2021*), suggesting that the enzyme may serve apicomplexan-specific, Ca$^{2+}$-responsive functions in remodeling the parasite phosphoproteome.

To track localization during the *T. gondii* lytic cycle, we tagged the endogenous C terminus of PP1 with an mNeonGreen (mNG) fluorophore and Ty epitope. We confirmed the Ca$^{2+}$-dependent stability of PP1 by immunoblotting (*Figure 5B*). Live microscopy revealed a diverse array of PP1 localizations in parasites (*Figure 5C*). In accordance with imaging in *Plasmodium* (*Zeeshan et al., 2021*), PP1-mNG was distributed diffusely in the cytoplasm, as well as in foci resembling the nucleus, centrosome, and in some parasites, the periphery. These diverse localizations may arise from the association of PP1 with distinct regulatory subunits forming different functional holoenzymes, as characterized in metazoans (*Brautigan and Shenolikar, 2018*). To determine whether PP1 exhibits Ca$^{2+}$-dependent relocalization, we treated parasites with zaprinast and A23187. The PP1 signal intensity increased at the apical cap and pellicle following zaprinast stimulation (*Figure 5C* and *Video 1*). However, Ca$^{2+}$ ionophore

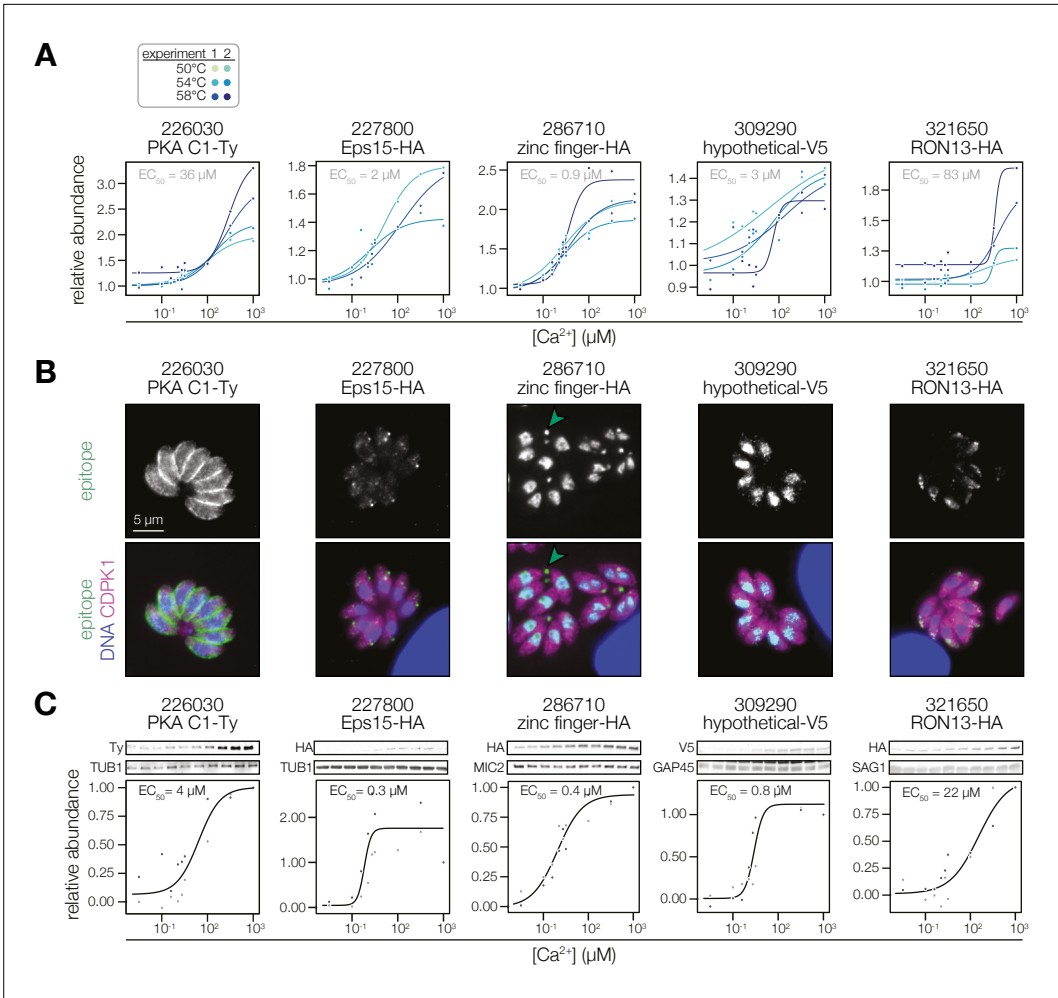

**Figure 4.** Validation of Ca$^{2+}$-dependent thermal stability. (**A**) Mass spectrometry-derived thermal profiles of the candidates, as in *Figure 3A*. (**B**) Immunofluorescence images of fixed intracellular parasites expressing the indicated proteins with C-terminal epitopes at endogenous loci. Hoechst and anti-CDPK1 were used as counterstains in the merged image. Green arrowheads highlight an example of TGGT1_286710 residual body staining. In the case of PKA C1/R, the stain of the R subunit is shown, as both subunits colocalize. (**C**) Immunoblot-derived thermal profiles of the candidates. Colors correspond to two independent replicates. Uncropped blots are shown in the *Figure 4—figure supplement 1*.

The online version of this article includes the following source data and figure supplement(s) for figure 4:

**Source data 1.** This file contains the source data that was quantified to make the graph presented in *Figure 4*.

**Source data 2.** This file contains the source data that was quantified to make the graph presented in *Figure 4*.

**Source data 3.** This file contains the source data that was quantified to make the graph presented in *Figure 4*.

**Source data 4.** This file contains the source data that was quantified to make the graph presented in *Figure 4*.

**Source data 5.** This file contains the source data that was quantified to make the graph presented in *Figure 4*.

**Source data 6.** This file contains the source data that was quantified to make the graph presented in *Figure 4*.

**Source data 7.** This file contains the source data that was quantified to make the graph presented in *Figure 4*.

**Source data 8.** This file contains the source data that was quantified to make the graph presented in *Figure 4*.

**Source data 9.** This file contains the source data that was quantified to make the graph presented in *Figure 4*.

**Source data 10.** This file contains the source data that was quantified to make the graph presented in *Figure 4*.

**Source data 11.** This file contains the annotated source data that was quantified to make the graph presented in *Figure 4*.

**Figure supplement 1.** Uncropped immunoblots, as in *Figure 4C*.

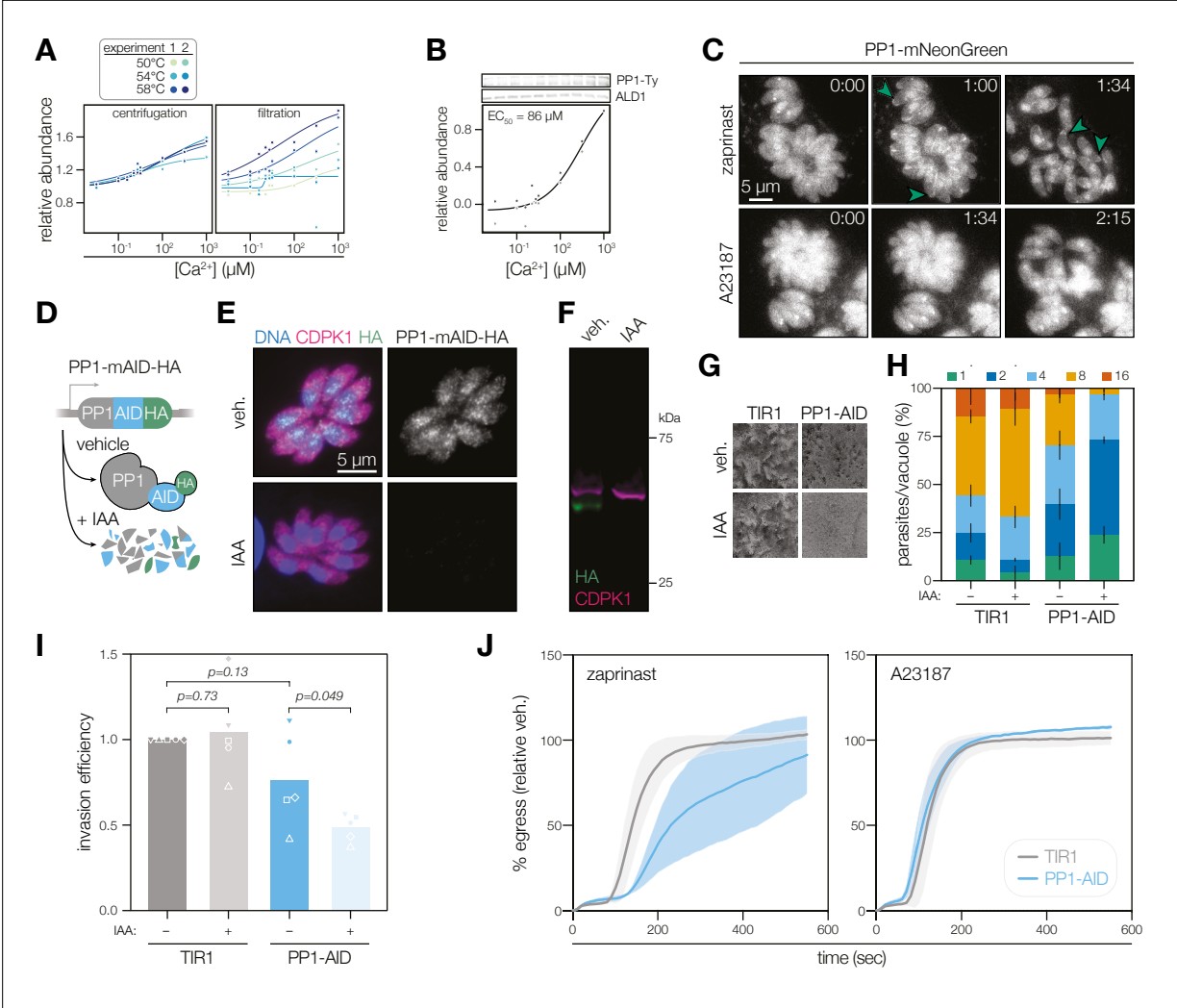

**Figure 5.** PP1 is a Ca²⁺-responsive enzyme involved in *T. gondii* egress and invasion. (**A**) The Ca²⁺-dependent stabilization of PP1 (TGGT1_310700) in each mass spectrometry experiment. (**B**) Immunoblotting for endogenously tagged PP1-mNG-Ty at different Ca²⁺ concentrations and thermal challenge at 58 °C. Abundance is calculated relative to the band intensity at 0 μM Ca²⁺ and scaled. Points in shades of gray represent different replicates. A dose-response curve was calculated for the mean abundances. (**C**) Parasites expressing endogenously tagged PP1-mNG egress after treatment with 500 μM zaprinast or 4 μM A23187. Arrows show examples of PP1 enrichment at the apical end. Time after treatment is indicated in m:ss. (**D**) Rapid regulation of PP1 by endogenous tagging with mAID-HA. IAA, Indole-3-acetic acid (IAA). (**E**) PP1-mAID-HA visualized in fixed intracellular parasites by immunofluorescence after 3 hr of 500 μM IAA or vehicle treatment. Hoechst and anti-CDPK1 are used as counterstains (*Waldman et al., 2020*). (**F**) PP1-mAID-HA depletion, as described in (**E**), monitored by immunoblotting. The expected molecular weights of PP1-mAID-HA and CDPK1 are 48 and 65 kDa, respectively. (**G**) Plaque assays of 1,000 TIR1 and PP1-mAID-HA parasites infected onto a host cell monolayer and allowed to undergo repeated cycles of invasion, replication, and lysis for 7 days in media with or without IAA. (**H**) The number of parasites per vacuole measured for PP1-mAID-HA and the TIR1 parental strain after 24 hr of 500 μM IAA treatment. Mean counts (n=3) are expressed as a percentage of all vacuoles counted. (**I**) Invasion assays PP1-mAID-HA or TIR1 parental strains treated with IAA or vehicle for 3 hr. Parasites were incubated on host cells for 60 min prior to differential staining of intracellular and extracellular parasites. Parasite numbers were normalized to host cell nuclei for each field. Means graphed for n=5 biological replicates (different shapes), Welch's t-test. (**J**) Parasite egress stimulated with 500 μM zaprinast or 8 μM A23187 following 3 h of treatment with vehicle or IAA. Egress was monitored by the number of host cell nuclei stained with DAPI over time and was normalized to egress in the vehicle-treated strain. Mean ±S.D. graphed for n=3 biological replicates.

The online version of this article includes the following source data for figure 5:

**Source data 1.** This file contains the source data that was quantified to make the graph presented in *Figure 4*.

**Source data 2.** This file contains the source data that was quantified to make the graph presented in *Figure 4*.

**Source data 3.** This file contains the annotated source data that was quantified to make the graph presented in *Figure 5*.

**Source data 4.** This file contains the source data that was presented in *Figure 5*.

**Source data 5.** This file contains the annotate source data that was presented in *Figure 5*.

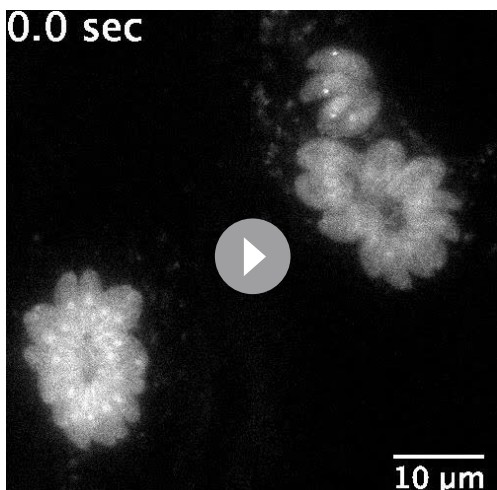

**Video 1.** Representative image series of parasites expressing endogenously tagged PP1-mNG following treatment with 500 μM zaprinast.

https://elifesciences.org/articles/80336/figures#video1

treatment failed to induce the same relocalization patterns (*Figure 5C* and *Video 2*). The dynamics of PP1 activity thus appear enhanced by cGMP signaling within the parasites, which is upstream of $Ca^{2+}$ (*Brown et al., 2016*; *Sidik et al., 2016b*) and was similarly observed in *Plasmodium* (*Paul et al., 2020*).

The parasite apex and pellicle are hotspots for the signaling that potentiates motility and invasion, so the relocalization of PP1 suggests it may function in these processes. We created strains expressing the PP1 catalytic subunit with an endogenous C-terminal auxin-inducible degron (AID) for rapid conditional knockdown (*Figure 5D*). Conditional degradation of PP1, following treatment with indole-3-acetic acid (IAA) was confirmed by immunofluorescence and immunoblotting (*Figure 5E and F*). Parasites depleted of PP1 failed to form plaques (*Figure 5G*), implicating the phosphatase in one or more essential functions during the lytic cycle (*Sidik et al., 2016a*). Even in the absence of IAA, the PP1-AID strain formed small plaques, indicating substantial hypomorphism. PP1-AID parasites exhibited slower replication than untagged parasites, and this effect was exacerbated with IAA treatment (*Figure 5H*). Parasites depleted of PP1 had a reduced invasion efficiency (*Figure 5I*), although this effect was modest and subject to technical variation, likely arising from the hypomorphism of the C-terminally tagged strain. Parasites treated with IAA egressed more slowly than untreated parasites when stimulated with zaprinast; however, egress kinetics were indistinguishable with a $Ca^{2+}$ ionophore agonist (*Figure 5J*). Together, these results suggest that PP1 holoenzymes function at multiple steps in the lytic cycle. At least one holoenzyme relocalizes when parasite cGMP/$Ca^{2+}$ pathways are stimulated. Because parasites lacking PP1 exhibit delays specifically in zaprinast-induced egress, we hypothesize that the peripheral holoenzyme enhances $Ca^{2+}$ signaling.

## A PP1 holoenzyme dephosphorylates signaling enzymes at the parasite periphery

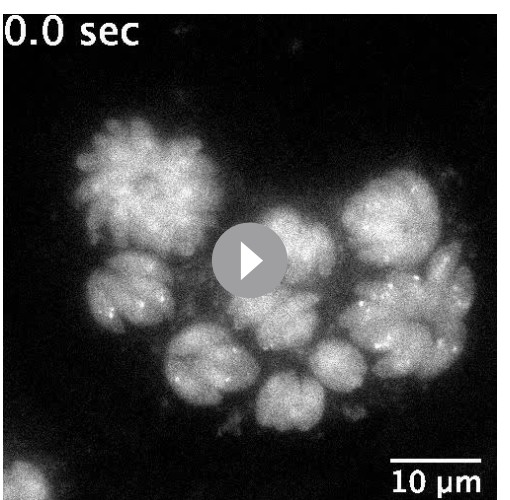

**Video 2.** Representative image series of parasites expressing endogenously tagged PP1-mNG following treatment with 4 μM A23187.

https://elifesciences.org/articles/80336/figures#video2

PP1 is one of the major serine/threonine phosphatases in eukaryotic cells (*Brautigan and Shenolikar, 2018*). As the catalytic subunit relocalizes during cGMP/$Ca^{2+}$-stimulated transitions in apicomplexans, we hypothesized that PP1 dephosphorylates crucial targets during egress and motility. To identify putative PP1 holoenzyme targets, we first treated intracellular PP1-AID parasites with IAA or vehicle for 3 hr prior to mechanically releasing parasites for analysis. We performed sub-minute phosphoproteomics by resuspending the extracellular parasites in a zaprinast solution followed by denaturation in SDS to stop enzymatic activity after 0, 10, 30, and 60 s. Multiplexing with TMTpro reagents followed by phosphopeptide enrichment allowed us to compare the zaprinast time courses with or without PP1-depletion in biological duplicate (*Figure 6A*). Analysis of an unenriched fraction of the proteome revealed significant depletion of

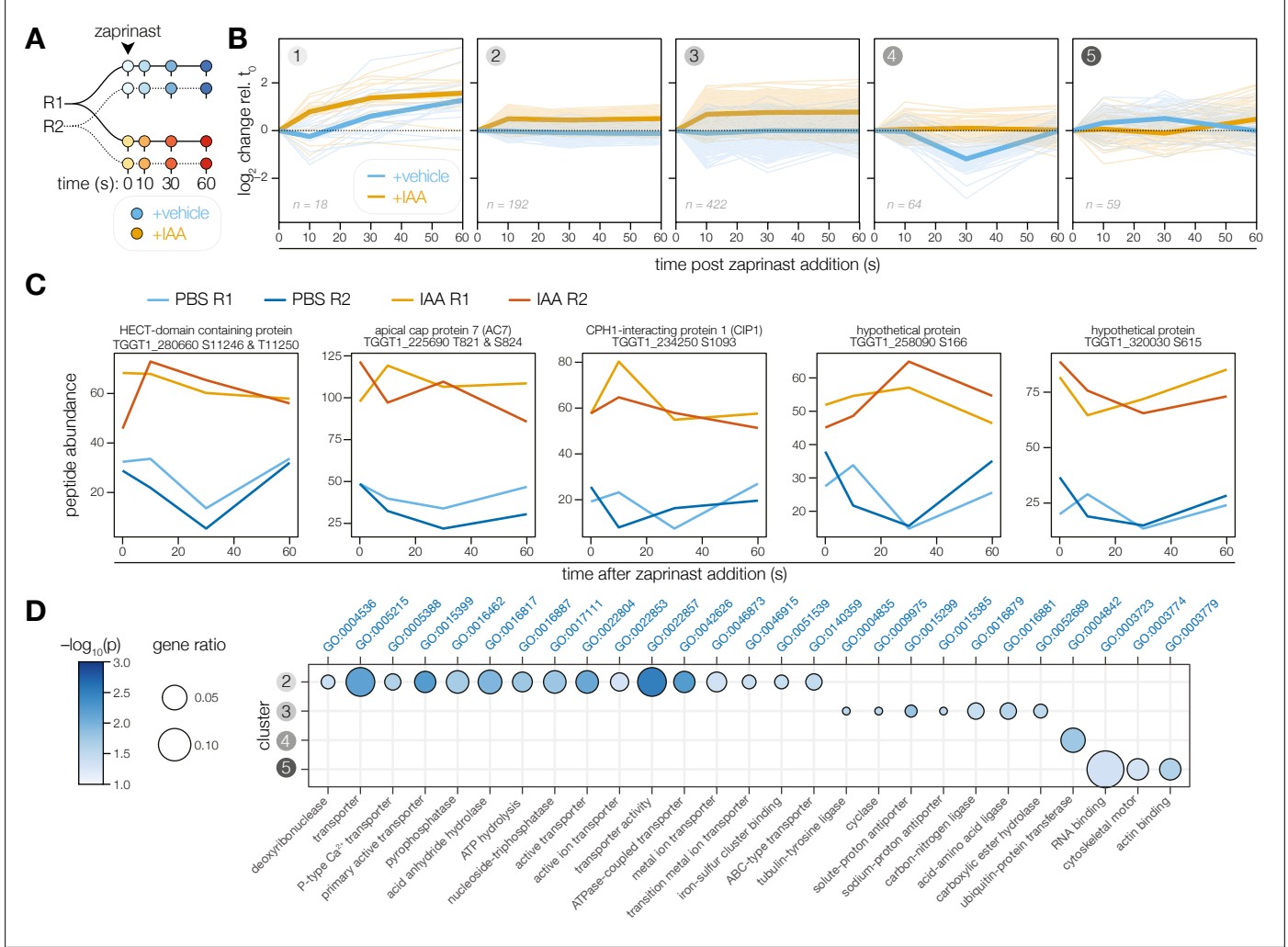

**Figure 6.** The PP1-dependent phosphoproteome. (**A**) Schematic of the phosphoproteomics time course. PP1-AID parasites were treated with IAA or vehicle for 3 hr. Extracellular parasites were then treated with zaprinast, and samples were collected during the first minute after stimulation. The experiment was performed in biological replicate (**R1 and R2**). (**B**) Five clusters were identified with respect to phosphopeptide dynamics and PP1-dependence. (**C**) Examples of phosphopeptides dynamically regulated by zaprinast and exhibiting PP1-dependent dephosphorylation. (**D**) GO terms enriched among phosphopeptides, grouped by cluster. Gene ratio is the proportion of proteins with the indicated GO term divided by the total number of proteins belonging to each cluster. Significance was determined with a hypergeometric test; only GO terms with p<0.05 and represented by more than one protein are shown. Redundant GO terms were removed. Cluster 1 lacked enough peptides for enrichment analysis.

The online version of this article includes the following source data and figure supplement(s) for figure 6:

**Figure supplement 1.** Extended analysis of PP1 phosphoproteomics experiment.

**Figure supplement 1—source data 1.** This file contains the source data that was presented in *Figure 6—figure supplement 1*.

**Figure supplement 1—source data 2.** This file contains the source data that was presented in *Figure 6—figure supplement 1*.

**Figure supplement 1—source data 3.** This file contains the annotated source data that was presented in *Figure 6—figure supplement 1*.

PP1, which we confirmed in parallel by immunoblot (*Figure 6—figure supplement 1*). The hypomorphism of the PP1-AID strain and reduced parasite yield resulted in a phosphoproteome with lower coverage: 6,916 phosphopeptides with quantification values. To identify peptides exhibiting PP1-dependent regulation, we selected those exhibiting a twofold difference in abundance between vehicle- and IAA-treated samples in both replicates for at least one time point. In total, 757 peptides passed this threshold.

To focus on zaprinast-dependent changes, we clustered these peptides on the basis of their abundance relative to the earliest time point of stimulation. The peptides fit into five clusters with respect

to PP1-dependence and kinetics (*Figure 6B*). Clusters 1, 2, and 3 contained peptides increasing in abundance upon stimulation in PP1-depleted parasites. The peptides belonging to cluster 1 generally increase in abundance with zaprinast treatment, which occurs more rapidly with PP1 depletion. By contrast, the abundances of the 614 peptides belonging to clusters 2 and 3 were elevated in the absence of PP1, suggesting that PP1 antagonizes these phosphorylation events. Under normal conditions, peptides in cluster 4 decreased sharply in abundance between 10 and 30 s of zaprinast treatment and recovered by 60 s; however, these peptides did not change in abundance when PP1 was depleted. Peptides in cluster 5 increased gradually between 10 and 30 s and decreased to basal levels by 60 s; when PP1 is depleted, these peptides exhibit a delay.

The effect of PP1 disruption on the phosphoproteome was pervasive, which may reflect disruption of numerous PP1 holoenzymes. Many of the examined phosphopeptides exhibited substantial basal hyperphosphorylation in the absence of PP1 (*Figure 6C*). Given the likely pleiotropy of catalytic subunit depletion, we focused our analysis on perturbed pathways rather than individual targets (*Figure 6D*). Cluster 1 did not possess enough peptides for enrichment analysis. Cluster 2 was enriched in phosphoproteins functioning in transmembrane transport, including P-type ATPases and ABC transporters. Cluster 3 contained both the guanylyl and adenylyl cyclases and was further enriched in putative sodium-hydrogen exchangers and tubulin-tyrosine ligases. Cluster 4 was solely enriched for proteins involved in ubiquitin transfer, including TGGT1_280660, an uncharacterized HECT domain-containing protein (*Figure 6C*). Cluster 5 phosphoproteins were notably involved in cytoskeletal motor activity, actin binding, and RNA binding. Numerous apical proteins exhibited PP1-dependent phosphorylation (*Table 1*), including AC7, CIP1, and two hypothetical proteins—TGGT1_258090 and TGGT1_320030, localized to the conoid base and the second apical polar ring, respectively (*Koreny et al., 2021*). The protein abundances did not vary between vehicle and IAA treatment, indicating that the phosphopeptide abundance changes arose from dynamic covalent modifications. The PP1-dependent phosphoproteome therefore supports the existence of apical and peripheral PP1 holoenzymes, as observed by live microscopy.

## PP1 activity is important for Ca²⁺ entry to enhance the kinetics of egress

The phosphoproteomics data pointed to ion homeostasis dysregulation in the absence of PP1 (*Figure 6D*). Numerous transporters are hyperphosphorylated when PP1 is depleted, including Ca²⁺ ATPases, proton transporters, and MFS and ABC-family transporters. We hypothesized that aberrant Ca²⁺ mobilization may underlie the phenotypes observed in the motile stages of the lytic cycle upon PP1 depletion (*Figure 5*), as Ca²⁺ release and signaling are required for egress and invasion. To monitor Ca²⁺ mobilization in parasites, we introduced the genetically encoded Ca²⁺ indicator GCaMP6f into a defined chromosomal locus (*Chen et al., 2013*; *Herneisen et al., 2020*). By tracking the vacuole fluorescence of PP1-depleted parasites, we observed delayed Ca²⁺ mobilization and egress following zaprinast treatment (*Figure 7A–C*, *Figure 7—figure supplement 1*, and *Video 3*). Despite the delay, IAA-treated parasites attained the same increase in Ca²⁺ levels prior to egress as vehicle-treated parasites (*Figure 7—figure supplement 1*). By contrast, Ca²⁺ mobilization upon A23187 treatment was unperturbed in PP1-depleted parasites (*Figure 7D–F*, *Figure 7—figure supplement 1* and *Video 4*). Motivated by recent experiments suggesting that *T. gondii* exhibits Ca²⁺-activated Ca²⁺ entry (*Vella et al., 2021*), we loaded PP1-AID parasites with the ratiometric Ca²⁺ indicator Fura2-AM and observed both reduced Ca²⁺ entry and lowered resting cytoplasmic [Ca²⁺] (*Figure 7G–I* and *Figure 7—figure supplement 1*). The integration of our global datasets therefore identified PP1 as a Ca²⁺-responsive enzyme and precipitated discovery of its role in Ca²⁺ entry.

## Discussion

Apicomplexan signaling pathways have largely been characterized through a combination of genetic manipulation, pharmacological perturbation, and physiological observation (*Bisio and Soldati-Favre, 2019*; *Brown et al., 2020*; *Lourido and Moreno, 2015*). Such approaches have been sufficient to document feedback loops between cyclic nucleotide signaling and Ca²⁺-store release and influx. However, the molecular mechanisms linking these pathways have remained obscure, as genomic annotation or bioinformatic analysis are complicated by the evolutionary divergence of apicomplexans. Unbiased,

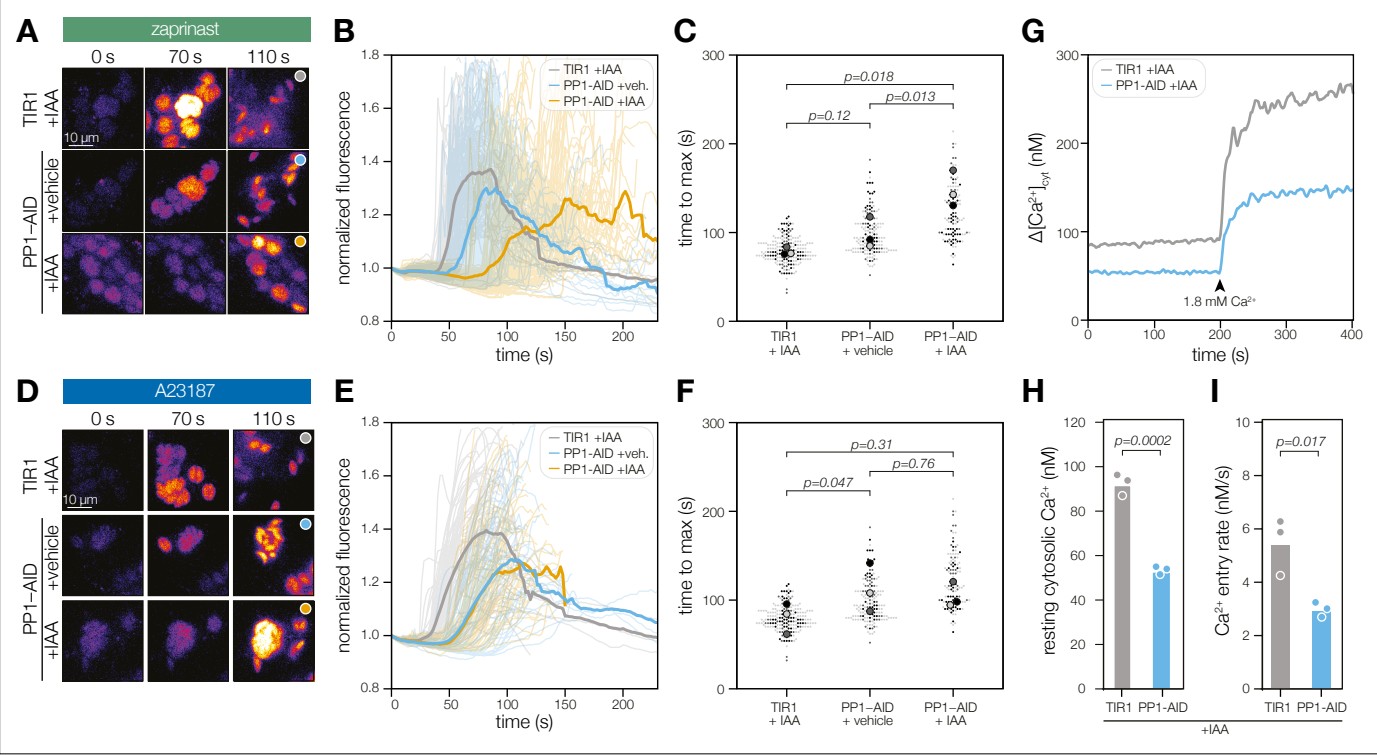

**Figure 7.** PP1 ensures rapid $Ca^{2+}$ mobilization prior to zaprinast-induced egress. (**A**) Selected frames of time-lapse images of PP1-AID parasites expressing the genetically encoded $Ca^{2+}$ indicator GCaMP following treatment with 500 μM zaprinast. (**B**) Normalized GCaMP fluorescence of individual vacuoles was tracked after zaprinast treatment and prior to egress (opaque lines). The solid line represents the mean normalized fluorescence of all vacuoles across n=3 biological replicates. (**C**) The time to maximum normalized fluorescence of individual vacuoles after zaprinast treatment. Different replicates are shown in different shades of gray. Small points correspond to individual vacuoles; large points are the mean for each replicate. p Values were calculated from a two-tailed *t*-test. (**D**) Selected frames of time-lapse images of PP1-AID parasites expressing the genetically encoded $Ca^{2+}$ indicator GCaMP following treatment with 4 μM A23187. (**E**) Normalized GCaMP fluorescence of individual vacuoles was tracked after A23187 treatment and prior to egress (opaque lines). The solid line represents the mean normalized fluorescence of all vacuoles across n=3 biological replicates. (**F**) The time to maximum normalized fluorescence of individual vacuoles after A23187 treatment. Small points correspond to individual vacuoles; large points are the mean for each replicate. p Values were calculated from a two-tailed *t*-test. (**G**) Fluorescence intensity of Fura2/AM-loaded TIR1 or PP1-AID parasites treated with IAA for 5 hr before and after addition of the 1.8 mM $Ca^{2+}$. Representative traces from three biological replicates. (**H**) Resting cytoplasmic [$Ca^{2+}$] prior to incubation in buffers with elevated [$Ca^{2+}$]. p Values were calculated from an ANOVA. (**I**) The rate of $Ca^{2+}$ entry in the first 20 s after addition of 1.8 mM $Ca^{2+}$ to parasites p values were calculated from an ANOVA. Entry rates following addition of other concentrations are shown in *Figure 7—figure supplement 1*.

The online version of this article includes the following figure supplement(s) for figure 7:

**Figure supplement 1.** PP1-AID parasites egress, as quantified by video microscopy.

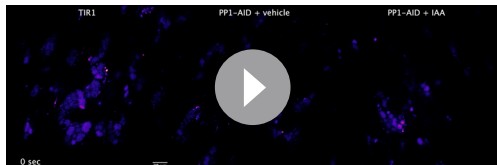

**Video 3.** Representative image series of PP1-AID parasites expressing the genetically encoded $Ca^{2+}$ indicator GCaMP following treatment with 500 μM zaprinast.

https://elifesciences.org/articles/80336/figures#video3

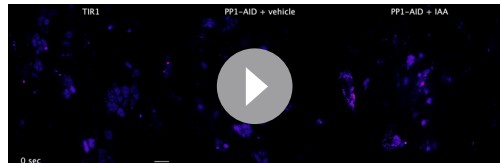

**Video 4.** Representative image series of PP1-AID parasites expressing the genetically encoded $Ca^{2+}$ indicator GCaMP following treatment with 4 μM A23187.

https://elifesciences.org/articles/80336/figures#video4

functional proteomics approaches have allowed us to map a Ca$^{2+}$-responsive proteome de novo. Our resource places proteins already under study within Ca$^{2+}$ signaling pathways and identifies hubs serving as pressure points for the perturbation of multiple signaling processes.

Our time-resolved phosphoproteomic data adds molecular depth to known changes in parasite physiology. Through the use of zaprinast we stimulate the endogenous regulation of Ca$^{2+}$ stores by PKG. Our phosphoproteomic analysis therefore includes Ca$^{2+}$-responsive pathways, as well as any other pathways downstream of cGMP or PKG, which may include Ca$^{2+}$-independent processes ( *Bullen et al., 2019*). The first cluster of phosphoproteins to respond to zaprinast stimulation includes second messenger amplifiers: cyclic nucleotide phosphodiesterases (PDE1 and PDE2); lipid kinases, phosphatases, and phospholipases; and the guanylyl cyclase (*Bisio et al., 2019*; *Brown and Sibley, 2018*). This cluster also contains early Ca$^{2+}$ sensors and transducers, such as *Tg*CDPK2A and *Tg*CDPK7 (*Bansal et al., 2021*), which additionally responded to Ca$^{2+}$ in our thermal profiling experiments. The next two clusters reveal rewiring of small molecule homeostasis, with non-Ca$^{2+}$ channels and transporters phosphorylated first. Parasites have long been reported to respond to extracellular H$^+$ and K$^+$ levels (*Moudy et al., 2001*; *Roiko and Carruthers, 2013*), although the proteins responsible and sequence of events transducing signals across the parasite plasma membrane remained obscure. Our proteomic datasets identified dynamic phosphorylation of ATPase2, NHE1, and two uncharacterized putative Ca$^{2+}$-activated K$^+$ channels. ATPase2 was recently reported to mediate essential lipid homeostasis at the parasite plasma membrane (*Chen et al., 2021*). Mutants lacking NHE1 exhibited elevated cytosolic Ca$^{2+}$, yet were less susceptible to Ca$^{2+}$ ionophore-induced egress (*Arrizabalaga et al., 2004*), suggesting that monovalent cation and Ca$^{2+}$ homeostasis mechanisms are intertwined. Indeed, the third wave of phosphorylation targets Ca$^{2+}$ and other divalent cation ATPases, including *Tg*A1, *Tg*CuTP, and other P-type ATPases with uncharacterized ion specificity. Transporters belonging to these clusters are noted *Tg*CDPK3 substrates, including ATP4, SCE1, and ApiAT5-3 (*Treeck et al., 2014*; *McCoy et al., 2017*; *Dominicus et al., 2021*); disruption of these transporters failed to alter cytosolic Ca$^{2+}$ levels (*McCoy et al., 2017*; *Parker et al., 2019*; *Wallbank et al., 2019*). A cluster of dynamic dephosphorylation events highlights the need for phosphatases to reverse modifications as parasites transition from one phase to another (*Brautigan and Shenolikar, 2018*; *Yang and Arrizabalaga, 2017*). These phosphoproteins were also enriched for channel or transporter functions, and in several cases also contained sites regulated by kinases. Apicomplexans must maintain homeostasis as ion gradients are inverted when parasites transition from intracellular to extracellular, likely requiring extreme rewiring of cellular physiology. Our phosphoproteomics dataset reveals that parasite ion homeostasis pathways are hard-wired into second messenger signal transduction, perhaps in anticipation of environmental challenge.

We can infer mechanisms of crosstalk between second messenger signaling networks, different classes of enzymes, and post-translational modifications from our global proteomic profiles. Proteins regulated by several kinases and phosphatases with different kinetics may function to concentrate these enzymes in local signaling hotspots acting as signaling scaffolds. Several of these dynamically regulated proteins themselves contain second messenger–sensing domains, such as EF hands (TGGT1_216620 and *Tg*CDPK2A) and cyclic nucleotide–binding domains (PDE2 and AAP2). Intriguingly, both subunits of PKA, the best-characterized cAMP receptor in *T. gondii*, are stabilized by Ca$^{2+}$. PKA has been proposed to negatively regulate Ca$^{2+}$ signaling in parasites (*Jia et al., 2017*; *Uboldi et al., 2018*). Here, we additionally observe Ca$^{2+}$ regulating PKA, although the molecular basis of this stability change remains to be established. Finally, the stimulated phosphoproteome exhibits extensive crosstalk with GTPase regulation and ubiquitin transfer pathways, suggesting that there remain vast and unexplored signaling networks, in addition to phosphorylation, that remodel cellular states in *T. gondii*.

The Ca$^{2+}$ responsiveness of PP1 revealed by our TPP analysis was surprising. In most organisms, the dominant Ca$^{2+}$-regulated phosphatase is calcineurin (*Shi, 2009*). Although PP1 holoenzymes have been reported to modulate Ca$^{2+}$ channels, the Ca$^{2+}$-responsive behavior of PP1 has not been reported in any other eukaryote. The function of the phosphatase had not been directly studied in *T. gondii*, although PP1 inhibitors have been shown to block invasion of host cells (*Delorme et al., 2002*) and recent experiments in *P. falciparum* suggest that PP1 is required for an as-yet undetermined step in the kinetic phase of infection (*Paul et al., 2020*). Guided by our global experiments, we endogenously tagged the C terminus of PP1 and observed diverse localization patterns. Numerous regulatory

subunits control the activity, localization, and specificity of the single catalytic subunit of the phosphatase (*Yang and Arrizabalaga, 2017*), giving rise to functionally distinct PP1 holoenzymes that play conserved roles in cell cycle progression (*Bertolotti, 2018*). Stimulating the transition from the replicative to the kinetic phase by treating parasites with the phosphodiesterase inhibitor zaprinast led to rapid enrichment of PP1 at the parasite periphery and apical end, hotspots of parasite signaling. Notably, PP1 was recently reported to relocalize to the apical complex in *Plasmodium* berghei ookinetes (*Zeeshan et al., 2021*). Although the proposed role of PP1 in ookinetes was to set cell polarity, given our phenotypic observations in *Toxoplasma*, we propose instead that the apical PP1 holoenzyme serves apicomplexan-specific, $Ca^{2+}$-responsive functions supporting the parasitic lifestyle (*Park et al., 2019*; *Paul et al., 2015*; *Philip and Waters, 2015*). Indeed, parasites conditionally depleted of PP1 exhibited reduced invasion and delayed host cell lysis in response to zaprinast.

Many parasite PP1 holoenzymes are as yet undefined. A recent phosphoproteome of *P. falciparum* blood-stage parasites depleted of PP1 identified ~5,000 phosphosites from ~1,000 proteins (*Paul et al., 2020*). Despite complications in determining direct substrates, the study proposed PP1 regulation of a HECT E3 ligase and guanylate cyclase upstream of PKG, providing a potential functional link between PP1 and motility. Our experimental design benefited from higher temporal resolution and the ability to directly trigger the signaling pathways in question, providing a more comprehensive analysis of phosphatase activity during stimulated motility. We uncovered two patterns of PP1-dependent regulation: (i) sites hyperphosphorylated only in the absence of PP1 (clusters 2 and 3), and (ii) zaprinast-dependent sites no longer regulated in the absence of PP1 (cluster 4). The guanylyl cyclase and possible ortholog of the HECT E3 ligase identified in the *P. falciparum* study both included PP1-dependent phosphosites in our experiment, suggesting that these PP1-regulated cascades are important and conserved in apicomplexan parasites. The guanylyl cyclase belongs to one of the hyperphosphorylated clusters. If guanylyl cyclase function is compromised in the absence of PP1, accumulation of cGMP to the levels required for egress may be delayed following inhibition of phosphodiesterases, partially explaining the delayed zaprinast response observed in PP1-depleted parasites (*Paul et al., 2020*). Additional work probing the cGMP pathway—for example by directly monitoring cyclic nucleotide levels or leveraging mutants expressing conditional depletion alleles of phosphodiesterases, cyclases, and kinases—is required to precisely place PP1 in the network that primes parasites for the kinetic phase of the infection cycle. Moreover, the set of phosphoproteins normally dephosphorylated by PP1 during zaprinast treatment was uniquely enriched in ubiquitin-transfer enzymes. Prior proteomic enrichment studies observed an abundance of ubiquitin modifications in cell-cycle-regulated proteins, including the peripheral parasite inner membrane complex (*Silmon de Monerri et al., 2015*). However, the ubiquitination pathway is largely unexplored in the context of apicomplexan motility. HECT E3 ligase inhibitors reduced merozoite egress in *P. falciparum* (*Paul et al., 2020*). Our proteomic datasets reveal that the phosphorylation states of ubiquitin transferases and hydrolases change within 60 s of cGMP elevation, suggesting that dynamic ubiquitination intersects with dynamic phosphorylation to regulate transitions between intracellular and extracellular states.

The enrichment of transporters in the basally dysregulated sites motivated us to explore PP1 phenotypes associated with $Ca^{2+}$ mobilization and entry. We observed that PP1 activity enhances $Ca^{2+}$ entry, which itself was reported to enhance parasite motility (*Pace et al., 2014*; *Vella et al., 2021*). Synthesizing the phosphoproteomic, phenotypic, and thermal profiling data, which reports a micromolar-range $EC_{50}$ of PP1 for $Ca^{2+}$, we propose that the $Ca^{2+}$-regulated PP1 holoenzyme assembles following an initial wave of $Ca^{2+}$ store release, and then inhibits $Ca^{2+}$ uptake, thereby contributing to the overall elevation of cytoplasmic $Ca^{2+}$. The identification of specific PP1 holoenzymes and genetic manipulation of regulatory subunits is required to test this model; however, such negative feedback loops between $Ca^{2+}$-stimulated kinases and $Ca^{2+}$ uptake have been reported for *Tg*CDPK1 and *Tg*CDPK3 (*Herneisen et al., 2020*; *Stewart et al., 2017*). Alternatively, PP1 itself may change membrane association or structure. The $Ca^{2+}$ signaling field has benefited from decades of research in other model organisms, in which pathway connections were incrementally extended based on prior knowledge about the system (*Berridge et al., 2000*; *Clapham, 2007*; *Luan and Wang, 2021*). By contrast, we developed a global mass spectrometry-based approach to systematically identify the protein components of $Ca^{2+}$ signaling pathways on the basis of dynamic phosphoregulation and biochemical thermal stability. We applied this technology to apicomplexans, which are widespread eukaryotic human pathogens

whose signaling pathways remain largely unmapped due to their evolutionary divergence from model organisms (*Lourido and Moreno, 2015*). Prior proteomic surveys investigating the contribution of individual kinases (*Alam et al., 2015*; *Balestra et al., 2021*; *Brochet et al., 2014*; *Jia et al., 2017*; *Patel et al., 2019*; *Treeck et al., 2014*) or phosphatases (*Paul et al., 2020*; *Yang et al., 2019*) to global phosphoregulation lacked the temporal resolution to map transitions between the replicative and kinetic phases of the apicomplexan infection cycle, which occur in a matter of seconds. Advances in proteomics technologies, such as enhanced sensitivity and improved multiplexing methods, are now being leveraged to monitor the sub-minute processes of exflagellation and egress (*Dominicus et al., 2021*; *Invergo et al., 2017*). Second-messenger signaling pathways underpin these transitions at distinct developmental stages in diverse apicomplexans, with many kinases and phosphatases functioning at multiple steps in the parasitic life cycle. Meta-analyses of orthogonal proteomic profiles, such as a phosphoproteome tracking ionophore-induced egress (*Dominicus et al., 2021*), could reveal signatures of signaling subnetworks. A recent study points to extensive regulation of enzymes in the cyclic nucleotide pathways following treatment with calcium ionophore, which likely bypasses some of the regulated steps of calcium release and uptake from parasite stores—PDE1, PDE2, and the guanylyl and adenylyl cyclases were all heavily regulated (*Dominicus et al., 2021*). We identified some of these enzymes (PDE2, GC, and the CNB protein AAP2) as candidate signaling platforms, as in our study they belonged to several kinetically resolved clusters, suggesting regulation by multiple kinases or phosphatases. In principle, our approach can be applied to other post-translational modifications, natural ligands (*Lim et al., 2018*; *Sridharan et al., 2019*), and organisms (*Jarzab et al., 2020*) to establish—and in some cases re-evaluate—the topology of complex signaling pathways.

# Materials and methods

## Key resources table

| Reagent type (species) or resource | Designation | Source or reference | Identifiers | Additional information |
|---|---|---|---|---|
| Strain, strain background (*T. gondii*) | TIR1 | PMID:28465425 | | RH/TIR1/ΔKU80/ΔHXGPRT |
| Strain, strain background (*T. gondii*) | TIR1/GCaMP6f | PMID:35484233 | | RH/TIR1/pTUB1-GCaMP6f-3'DHFR/ΔKU80/ΔHXGPRT |
| Strain, strain background (*T. gondii*) | DiCre | PMID:31577230 | | RH/ΔKU80::DiCre_T2A/ΔHXGPRT |
| Strain, strain background (*T. gondii*) | TIR1/MIC2-GLuc-P2A-GCaMP6f | This paper | | RH/TIR1/ΔKU80/ΔHXGPRT/pMIC2-MIC2-GLuc-myc-P2A-GCaMP6f |
| Strain, strain background (*T. gondii*) | PKA C1-Ty (*Figure 4*) | This paper | TGGT1_226030 | RH/TIR1/pTUB1-GCaMP6f-3'DHFR/ΔKU80/PKA C1-mCherry-V5-mAID-Ty/PKA R-V5-3HA |
| Strain, strain background (*T. gondii*) | Eps15-HA (*Figure 4*) | This paper | TGGT1_227800 | RH/TIR1/pTUB1-GCaMP6f-3'DHFR/ΔKU80/Eps15-V5-mCherry-mAID-HA |
| Strain, strain background (*T. gondii*) | zinc finger-HA (*Figure 4*) | This paper | TGGT1_286710 | RH/TIR1/ΔKU80/ΔHXGPRT/TGGT1_286470-V5-3HA |
| Strain, strain background (*T. gondii*) | hypothetical-V5 (*Figure 4*) | This paper | TGGT1_309290 | RH/TIR1/pTUB1-GCaMP6f-3'DHFR/ΔKU80/ΔHXGPRT/TGGT1_309290-V5-mNeonGreen-mAID-Ty |
| Strain, strain background (*T. gondii*) | RON13-HA (*Figure 4*) | This paper | TGGT1_321650 | RH/ΔKU80::DiCre_T2A/ΔHXGPRT/RON13-HA-U1 |
| Strain, strain background (*T. gondii*) | PP1-AID (*Figure 5*) | This paper | TGGT1_310700 | RH/TIR1/ΔKU80/ΔHXGPRT/pMIC2-MIC2-GLuc-myc-P2A-GCaMP6f/PP1-V5-mAID-HA |
| Strain, strain background (*T. gondii*) | PP1-AID/TIR1/MIC2-GLuc-P2A-GCaMP6f (*Figure 7*) | This paper | TGGT1_310700 | RH/TIR1/ΔKU80/ΔHXGPRT/pMIC2-MIC2-GLuc-myc-P2A-GCaMP6f/PP1-V5-mAID-HA |
| Cell line (*Homo sapiens*) | Human Foreskin Fibroblasts (HFFs) | ATCC | SCRC-1041 | |
| Antibody | Guinea pig monoclonal anti-CDPK1 | Covance | Custom antibody | IF (1/10000), WB (1/40000) |

*Continued on next page*

*Continued*

| Reagent type (species) or resource | Designation | Source or reference | Identifiers | Additional information |
|---|---|---|---|---|
| Antibody | Mouse monoclonal anti-TUB1 (clone 12G10) | Developmental Studies Hybridoma Bank at the University of Iowa | RRID:AB_1157911 | WB (1/5000) |
| Antibody | Mouse monoclonal anti-Ty1 (clone BB2) | PMID:8813669 | | IF (1/1000), WB(1/2000) |
| Antibody | Rabbit polyclonal anti-HA | Invitrogen | Invitrogen:71–5500 | WB (1/1000) |
| Antibody | Mouse monoclonal anti-MIC2 (6D10) | PMID:10799515 | | WB (1:2000) |
| Antibody | Mouse monoclonal anti-V5 | Invitrogen | Invitrogen:R960-25 | IF (1:1000), WB (1:2000) |
| Antibody | Rabbit polyclonal | PMID:18312842 | | WB (1:2000) |
| Antibody | Mouse monoclonal anti-HA (16B12) | Biolegend | Biolegend:901533 | IF (1:1000) |
| Antibody | Mouse polyclonal anti-SAG1 | PMID:3183382 | | WB (1/1000) |
| Antibody | Alexa Fluor 594 polyclonal goat anti-guinea pig | Life Technologies | Life Technologies:A11076 | IF (1/1000) |
| Antibody | Alexa Fluor 488 polyclonal goat anti-mouse | Life Technologies | Life Technologies:A11029 | IF (1/1000) |
| Antibody | IRDye 800CW polyclonal Goat anti-Mouse IgG1-Specific Secondary Antibody | LICOR | LICOR:926–32350 | WB (1/10000) |
| Antibody | IRDye 680LT polyclonal Goat anti-Mouse IgG Secondary Antibody | LICOR | LICOR:926–68020 | WB (1/10000) |
| Antibody | IRDye 800CW polyclonal Donkey anti-Guinea Pig IgG Secondary Antibody | LICOR | LICOR:926–32411 | WB (1/10000) |
| Antibody | IRDye 680RD polyclonal Donkey anti-Guinea Pig IgG Secondary Antibody | LICOR | LICOR:926–68077 | WB (1/10000) |
| Antibody | IRDye 800CW polyclonal Goat anti-Rabbit IgG Secondary Antibody | LICOR | LICOR:926–32211 | WB (1/10000) |
| Antibody | IRDye 680LT polyclonal Goat anti-Rabbit IgG Secondary Antibody | LICOR | LICOR:926–68021 | WB (1/10000) |
| Chemical compound, drug | Hoechst | Santa Cruz | Santa Cruz:sc-394039 | IF (1/20000) |
| Chemical compound, drug | Prolong Diamond | Thermo Fisher | Thermo Fisher:P36965 | |
| Chemical compound, drug | zaprinast | Calbiochem | Calbiochem:684500 | |
| Chemical compound, drug | A23187 | Calbiochem | Calbiochem:100105 | |
| Commercial assay or kit | S-trap micro | Protifi | Protific:C02-micro-80 | |
| Commercial assay or kit | TMT10plex Isobaric Label Reagent Set | Thermo Fisher Scientific | Thermo Fisher Scientific:90111 | |
| Commercial assay or kit | TMTpro 16plex Label Reagent Set | Thermo Fisher Scientific | Thermo Fisher Scientific:A44522 | |
| Commercial assay or kit | High-Select TiO2 Phosphopeptide Enrichment Kit | Thermo Fisher Scientific | Thermo Fisher Scientific:A32993 | |
| Commercial assay or kit | High-Select Fe-NTA Phosphopeptide Enrichment Kit | Thermo Fisher Scientific | Thermo Fisher Scientific:A32992 | |

*Continued on next page*

*Continued*

| Reagent type (species) or resource | Designation | Source or reference | Identifiers | Additional information |
|---|---|---|---|---|
| Commercial assay or kit | Pierce High pH Reversed-Phase Peptide Fractionation Kit | Thermo Fisher Scientific | Thermo Fisher Scientific:84868 | |
| Commercial assay or kit | Calcium Calibration Buffer Kit #1, zero and 10 mM CaEGTA | Life Technologies | Life Technologies:C3008MP | |
| Commercial assay or kit | Hydrophobic Sera-Mag Speed Beads | GE Healthcare | GE Healthcare:65152105050250 | |
| Commercial assay or kit | Hydrophilic Sera-Mag Speed Beads | GE Healthcare | GE Healthcare:45152105050250 | |
| Recombinant DNA reagent | All plasmids used in this study are listed in *Supplementary file 7* | | | |
| Sequence-based reagent | All primers and oligonucleotides used in this study are listed in *Supplementary file 7* | | | |
| Software, algorithm | Proteome Discoverer 4.2 | Thermo Fisher | | |
| Software, algorithm | R version 4.0 | R Foundation for Statistical Computing | | |
| Software, algorithm | mineCETSA version 1.1.1 | *Dziekan et al., 2020* | | https://github.com/nkdailingyun/mineCETSA |
| Software, algorithm | Prism 8 | GraphPad | | |
| Software, algorithm | HHPRED | PMID:29258817 | | |
| Other | Halt protease inhibitor | Thermo Fisher | Thermo Fisher:87786 | Materials and Methods: lysis buffer |
| Other | Halt protease and phosphatase inhibitor | Thermo Fisher | Thermo Fisher:PI78440 | Materials and Methods: lysis buffer |
| Other | Benzonase | Sigma Aldrich | Sigma Aldrich:E1014 | Materials and Methods: lysis buffer |

## Reagents and cell culture

*T. gondii* parasites of the type I RH strain Δku80/Δhxgprt genetic background (ATCC PRA-319) were grown in human foreskin fibroblasts (HFFs, ATCC SCRC-1041) maintained in DMEM (GIBCO) supplemented with 3% inactivated fetal calf serum (IFS) and 10 µg/mL gentamicin (Thermo Fisher Scientific). When noted, DMEM was supplemented with 10% IFS and 10 µg/mL gentamicin.

## Parasite transfection and strain construction

### Genetic background of parasite strains

Existing *T. gondii* RH strains were used as genetic backgrounds for this study. All strains contain the Δku80Δhxgprt mutations to facilitate homologous recombination (*Huynh and Carruthers, 2009*). TIR1/RH expresses the TIR1 ubiquitin ligase (*Brown et al., 2017*). TIR1/pTUB1-GCaMP additionally expresses a GCaMP6f transgene (*Smith et al., 2022*). DiCre_T2A/RH uses the same promoter to express both subunits of Cre, separated by T2A skip peptides (*Hunt et al., 2019*).

### TIR1/pMIC2-MIC2Gluc-P2A-GCaMP6f

A new TIR1/RH strain expressing GCaMP6f under different regulatory elements was constructed as a part of this study. The sequence pMIC2_MIC2Gluc_myc_P2A_GCaMP6f_DHFR3'UTR was amplified with primers P1 and P2 from plasmid Genbank ON312866 to yield a repair template with homology to the 5' and 3' ends of a defined, neutral genomic locus (*Markus et al., 2019*). Approximately 1×10^7 extracellular TIR1 parasites were transfected with 20 µg gRNA/Cas9-expression plasmid pBM041 (GenBank MN019116.1) and 10 µg repair template. GFP-positive clones were isolated by limiting dilution following fluorescence-activated cell sorting.

### Endogenous tagging of candidate genes

Genes were endogenously tagged using the previously described high-throughput (HiT) tagging strategy (*Smith et al., 2022*). Cutting units specific to each candidate were purchased as IDT gBlocks

(P3-P7) and assembled with the following empty HiT vector backbones: pALH083 (V5-mCherry-mAID-Ty; GenBank: ON312867), pALH052 (V5-mCherry-mAID-HA; GenBank: OM863784), pALH047 (V5-3HA; GenBank: ON312868), pGL015 (V5-mNG-mAID-Ty; GenBank OM640005), pEC341 (HA-loxP-U1; GenBank: OM863788), and pALH086 (V5-mAID-HA; GenBank ON312869).

Between 20 and 50 μg of each vector was BsaI-linearized and co-transfected with 50 μg of the pSS014 Cas9-expression plasmid (GenBank: OM640002). Vectors targeting TGGT1_226030 (PKA-C1) and TGGT1_227800 (Eps15) were transfected into TIR1/pTUB1-GCaMP or TIR1/pMIC2-MIC2Gluc-P2A-GCaMP6f. Vectors targeting TGGT1_286710 and TGGT1_309290 were transfected into TIR1/RH. Vectors targeting TGGT1_321650 were transfected into DiCre/RH. Parasite populations were selected for 1 week in standard media with 1 μM pyrimethamine or 25 μg/mL mycophenolic acid and 50 μg/mL xanthine, followed by subcloning into 96-well plates. Single clones were screened for tag expression by immunofluorescence and immunoblot.

## Endogenous tagging of PP1

A cutting unit specific to the C-terminus of PP1 (TGGT1_310700; P8) was assembled into the pALH086 HiT vector backbone. Approximately 50 μg of this vector was BsaI-linearized and co-transfected with 50 μg of the pSS014 Cas9-expression plasmid into TIR1/RH or Parasite populations were selected for 1 week in standard media with 25 μg/mL mycophenolic acid and 50 μg/mL xanthine and were then subcloned by limiting dilution. Single clones were screened for tag expression by immunofluorescence, immunoblot and sequencing of the junction spanning the 3' CDS and the tag.

## Sub-minute phosphoproteomics experiments

### Parasite harvest and treatment

*T. gondii* tachyzoites from the RH strain were infected onto confluent HFF monolayers in 4–8 15 cm dishes. After the parasites had completely lysed the host cell monolayer (40–48 hr post-infection), extracellular parasites were passed through 5 μm filters into 50 ml conical vials. The samples were spun at 1000 x g for 7 min. The supernatant was decanted, and the parasite pellet was resuspended in 1 ml FluoroBrite DMEM lacking serum and transferred to a 1.5 ml protein low-bind tube. The sample was spun in a mini centrifuge at 1000 x g for 7 min. The supernatant was aspirated, and the parasite pellet was resuspended in 800 μl FluoroBrite DMEM. The sample was split into aliquots of 300 and 500 μl, which were spun at 1000 x g for 7 min followed by aspiration of the supernatant. The pellet containing $\frac{5}{8}$ of the parasite harvest was resuspended in 250 μl 500 μM zaprinast in FluoroBrite DMEM. Aliquots of 50 μl were removed and combined with 50 μl 2 X lysis buffer (10% SDS, 4 mM $MgCl_2$, 100 mM TEAB pH 7.55 with 2 X Halt Protease and Phosphatase Inhibitors and 500 U/ml benzonase) at 0, 5, 10, 30, and 60 s post-stimulation. The final composition of the lysate was 5% SDS, 2 mM $MgCl_2$, 50 mM TEAB pH 8.5 with 1 X Halt Protease and Phosphatase Inhibitors and 250 U/ml benzonase. The pellet containing $\frac{3}{8}$ of the parasite harvest was resuspended in 150 μl 0.5% DMSO in FluoroBrite DMEM. Aliquots of 50 μl were removed and combined with 50 μl 2 X lysis buffer at 0, 10, and 30 s post-stimulation.

### Protein cleanup and digestion

Proteins were cleaned up and digested using a modified version of the S-trap protocol (Protifi). Proteins in the lysates were reduced with the addition of 5 mM TCEP and heating at 55 °C for 20 min. Alkylation was performed for 10 min at room temperature with 20 mM MMTS. The lysates were acidified to a final concentration of 2.5% v/v phosphoric acid. A 6 X volume of S-trap binding buffer (90% methanol, 100 mM TEAB pH 7.55) was added to each sample to precipitate proteins. The solution was loaded onto S-trap mini columns (Protifi) and spun at 4000 x g until all of the solution had been passed over the column. The columns were washed four times with 150 μl S-trap binding buffer, followed by a 30 s spin at 4000 x g. Proteins were digested overnight at 37 °C in a humidified incubator with 20 μl of 50 mM TEAB pH 8.5 containing 2 μg of trypsin/LysC mix (Thermo Fisher Scientific). Peptides were eluted in three 40 μl washes with 50 mM TEAB, 0.2% formic acid, and 50% acetonitrile/0.2% formic acid. The eluted peptides snap-frozen and lyophilized.

## TMTpro labeling

The dried peptides were resuspended in 50 µl 100 mM TEAB 8.5. The peptide concentrations of 1/50 dilutions of the samples were quantified using the Pierce Fluorometric Peptide Assay according to manufacturer's instructions. Sample abundances were normalized to 50 µg peptides in 50 µl 100 mM TEAB pH 8.5. Each sample was combined with TMTpro reagents at a 5:1 label:peptide weight/weight ratio. Labeling reactions proceeded for 1 hr at room temperature with shaking at 350 rpm. Unreacted TMT reagent was quenched with 0.2% hydroxylamine. The samples were pooled, acidified to 3% with formic acid, and were loaded onto an EasyPep Maxi Sample Prep column. The samples were washed and eluted according to the manufacturer's instructions. 5% by volume of the elution was reserved as an unenriched proteome sample. The eluted peptides were snap-frozen and lyophilized until dry.

## Phosphopeptide enrichment

Phosphopeptides were enriched using the SMOAC protocol (*Tsai et al., 2014*). Resuspended TMTpro-labeled samples were enriched with the High-Select TiO2 Phosphopeptide Enrichment Kit (Thermo Fisher Scientific A32993) according to the manufacturer's instructions. The flow-through and the eluate containing phosphopeptides were immediately snap-frozen and lyophilized. The flow-through was resuspended and enriched with the High-Select Fe-NTA Phosphopeptide Enrichment Kit (Thermo Fisher Scientific A32992) according to the manufacturer's instructions. The eluted phosphopeptides were immediately snap-frozen and lyophilized.

## Fractionation

Unenriched and enriched proteome samples were fractionated with the Pierce High pH Reversed-Phase Peptide Fractionation Kit according to the manufacturer's instructions for TMT-labeled peptides. The phosphopeptides enriched with the TiO2 and Fe-NTA kits were combined prior to fractionation.

## MS data acquisition

The fractions were lyophilized and resuspended in 10–20 µl of 0.1% formic acid for MS analysis and were analyzed on an Exploris 480 Orbitrap mass spectrometer equipped with a FAIMS Pro source (*Bekker-Jensen et al., 2020*) connected to an EASY-nLC chromatography system as described above. Peptides were separated at 300 nl/min on a gradient of 5–20% B for 110 min, 20–28% B for 10 min, 28–95% B for 10 min, 95% B for 10 min, and a seesaw gradient of 95–2% B for 2 min, 2% B for 2 min, 2–98% B for 2 min, 98% B for 2 min, 98–2% B for 2 min, and 2% B for 2 min. The orbitrap and FAIMS were operated in positive ion mode with a positive ion voltage of 1800 V; with an ion transfer tube temperature of 270 °C; using standard FAIMS resolution and compensation voltages of –50 and –65 V, an inner electrode temperature of 100 °C, and outer electrode temperature of 80 °C with 4.6 ml/min carrier gas. Full scan spectra were acquired in profile mode at a resolution of 60,000, with a scan range of 400–1400 m/z, automatically determined maximum fill time, 300% AGC target, intensity threshold of $5 \times 10^4$, 2–5 charge state, and dynamic exclusion of 30 s with a cycle time of 1.5 s between master scans. MS2 spectra were generated with a HCD collision energy of 32 at a resolution of 45,000 using TurboTMT settings with a first mass at 110 m/z, an isolation window of 0.7 m/z, 200% AGC target, and 120 ms injection time.

## Phosphoproteomics time course analysis

Raw files were analyzed in Proteome Discoverer 2.4 (Thermo Fisher Scientific) to generate peak lists and protein and peptide IDs using Sequest HT (Thermo Fisher Scientific) and the ToxoDB release49 GT1 protein database. The unenriched sample search included the following post-translational modifications: dynamic oxidation (+15.995 Da; M), dynamic acetylation (+42.011 Da; N-terminus), static TMTpro (+304.207 Da; any N-terminus), static TMTpro (+304.207 Da; K), and static methylthio (+45.988 Da; C). The enriched sample search included the same post-translational modifications, but with the addition of dynamic phosphorylation (+79.966 Da; S, T, Y). The mass spectrometry proteomics data have been deposited to the ProteomeXchange Consortium via the PRIDE partner repository (*Perez-Riverol et al., 2022*) with the dataset identifier PXD033765 and 10.6019/PXD033765. Protein and peptide abundances are reported in *Supplementary file 1* and *Supplementary file 2*, respectively.

Exported peptide and protein abundance files from Proteome Discoverer 2.4 were loaded into R (version 4.0.4). Summed abundances were calculated for both DMSO and zaprinast-treated samples,

incorporating the quantification values from both replicates. The ratio of zaprinast and DMSO peptide abundances was transformed into a modified Z score. Peptides with modified Z scores of 3 or above (839 peptides) or –1.8 or below (154 peptides) were considered dynamically changing. The normalized abundance values of peptides passing these thresholds were used as input for a principal component analysis using the R stats package (version 3.6.2). Clustering analysis was performed with the mclust package (*Scrucca et al., 2016*; version 5.4.7) using the $\log_2$ ratios of zaprinast-treated samples relative to the DMSO-treated t=0 samples. Cluster assignments are reported in *Supplementary file 3*.

### PP1-AID phosphoproteomics

Samples were prepared as described above, with the following modifications. PP1-AID parasites were infected onto confluent HFF monolayers. IAA or vehicle (PBS) was added to the cell culture media 29 hr post-infection. At 32 hr post-infection, the monolayers were scraped, and parasites were mechanically released from host cells by passage through a 27-gauge needle. The parasites were harvested and concentrated as described above. The time-course was initiated when the parasite pellet was resuspended in 250 µl of 500 µM zaprinast solution. Aliquots of 50 µl were removed and combined with 2 X lysis buffer at 0, 10, 30, and 60 s post-stimulation. The experiment was performed in biological duplicate, with vehicle- and IAA-treated parasites, for a total of 16 samples. The samples were prepared for mass spectrometry, labeled with TMTpro reagents, and enriched as described above. The mass spectrometry proteomics data have been deposited to the ProteomeXchange Consortium via the PRIDE partner repository (*Perez-Riverol et al., 2022*) with the dataset identifier PXD033765 and 10.6019/PXD033765.

Exported peptide and protein abundance files from Proteome Discoverer 2.4 were loaded into R (version 4.0.4). Peptides were clustered if they exhibited 2-fold difference in abundance between vehicle- and IAA-treated samples in both replicates for at least one time point. To focus on peptides dynamically changing during stimulation, we used for each peptide the grouped $\log_2$ ratios relative to the t=0 abundance for each treatment as input to the mclust package (*Scrucca et al., 2016*; version 5.4.7).

## Thermal profiling temperature range experiments

### Parasite harvest and treatment

Temperature range thermal profiling experiments were performed in biological duplicate on different days. We have optimized thermal profiling for *T. gondii* and have published detailed descriptions of the protocol (*Herneisen and Lourido, 2021*). Confluent HFF cells in 15 cm dishes were infected with $2–5 \times 10^7$ RH tachyzoites each. When the parasites had fully lysed the host cell monolayer (40–48 hr later), the extracellular parasites passed through a 5 µm filter. The parasite solution was concentrated by centrifugation for 10 min at 1000 x *g*. Parasites were resuspended in 1 ml of wash buffer (5 mM NaCl, 142 mM KCl, 1 mM MgCl$_2$, 5.6 mM glucose, 25 mM HEPES pH 7.2) and spun again for 10 min at 1000 x *g*.

The parasite pellet was resuspended in 1200 µl of lysis buffer (5 mM NaCl, 142 mM KCl, 1 mM MgCl$_2$, 5.6 mM glucose, 25 mM HEPES pH 7.2 with 0.8% IGEPAL CA-630, 1 X Halt Protease Inhibitors, and 250 U/ml benzonase) and subjected to three freeze-thaw cycles. Half of the lysate was combined with an equivalent amount of 2 X EGTA or 10 µM Ca$^{2+}$ solution. The parasite extracts were incubated at 37 °C/5% CO$_2$ for 5 min. The parasite suspension was aliquoted into PCR tubes and was heated at the following temperatures on two 48-well heat blocks: 37, 41, 43, 47, 50, 53, 56, 59, 63, and 67 °C. After 3 min, the tubes were removed from the heat blocks and were chilled on ice for 5 min. The lysates were transferred to a TLA-100 rotor and were spun at 100,000 x g for 20 min at 4 °C in a Beckman Ultra MAX benchtop ultracentrifuge. The solution, containing the soluble protein fraction, was removed for further processing.

### Protein cleanup and digestion

The concentrations of the 37 °C samples were determined with a DC Protein Assay (BioRad). A volume corresponding to 50 µg of the 37 °C sample was used for further analysis, and equivalent volumes were used from the remaining temperature range samples. Samples were prepared using a modified version of the SP3 protocol (*Hughes et al., 2019*). A total of 500 µg of a 1:1 mix of hydrophobic and hydrophilic beads (GE Healthcare 65152105050250 and 45152105050250) was added to each

sample, followed by a 6 X volume of 100% ethanol to induce aggregation. The samples were incubated at room temperature and 1000 rpm on a thermomixer for 30 min. The beads were magnetically separated, and the supernatant was removed. The beads were washed three times with 80% ethanol followed by magnetic separation. Each sample was resuspended in 88 µl reduction buffer (10 mM TCEP and 50 mM TEAB, pH 8.5) and heated at 55 °C for 1 hr. Samples were alkylated with 25 mM iodoacetamide shielded from light for 1 hr. 2 µg trypsin was added to each sample, and digestion proceeded overnight at 37 °C on a thermomixer shaking at 1000 rpm. The beads were magnetically separated, and the peptide eluatse were removed from further analysis.

## TMT10plex labeling
Peptides were labeled according to the manufacturer's protocol (Thermo Scientific 90111), with the following modifications. The eluates were combined with 200 µg of TMT10plex reagent in 41 µl of acetonitrile, for an estimated 1:4 w/w peptide:tag labeling reaction. The labeling proceeded for 1 hr at room temperature and was quenched for 15 min with 5% hydroxylamine. The samples were then pooled, flash-frozen, and lyophilized to dryness. The peptides were resuspended in 10% glacial acetic acid and were desalted using a Waters SepPak Light C18 cartridge according to the manufacturer's instructions.

## Fractionation
Samples were fractionated offline via reversed-phase high performance liquid chromatography. Samples were applied to a 10 cm ×2.1 mm column packed with 2.6 µm Aeris PEPTIDE XB-C18 media (Phenomenex) using 20 mM ammonium formate pH 10 in water as Buffer A, 100% acetonitrile as Buffer B, and Shimadzu LC-20AD pumps. The gradient was isocratic 1% Buffer A for 1 min at 150 µl/min with increasing Buffer B concentrations to 16.7% B at 20.5 min, 30% B at 31 min and 45% B at 36 min. Fractions were collected with a FRC-10A fraction collector and were pooled to eight samples per condition, followed by lyophilization.

## MS data acquisition
The fractions were lyophilized and resuspended in 10–20 µl of 0.1% formic acid for MS analysis and were analyzed on a Q-Exactive HF-X Orbitrap mass spectrometer connected to an EASY-nLC chromatography system using 0.1% formic acid as Buffer A and 80% acetonitrile/0.1% formic acid as Buffer B. Peptides were separated at 300 nl/min on a gradient of 6–9% B for 3 min, 9–31% B for 100 min, 31–75% B for 20 min, and 75 to 100% B over 15 min.

The orbitrap was operated in positive ion mode. Full scan spectra were acquired in profile mode with a scan range of 375–1400 m/z, resolution of 120,000, maximum fill time of 50ms, and AGC target of $3 \times 10^6$ with a 15 s dynamic exclusion window. Precursors were isolated with a 0.8 m/z window and fragmented with a NCE of 32. The top 20 MS2 spectra were acquired over a scan range of 350–1500 m/z with a resolution of 45,000, AGC target of $8 \times 10^3$, and maximum fill time of 100ms, and first fixed mass of 100 m/z.

## Thermal profiling temperature range analysis
Raw files were analyzed in Proteome Discoverer 2.4 (Thermo Fisher Scientific) to generate peak lists and protein and peptide IDs using Sequest HT (Thermo Fisher Scientific) and the ToxoDB release49 GT1 protein database. The search included the following post-translational modifications: dynamic phosphorylation (+79.966 Da; S, T, Y), dynamic oxidation (+15.995 Da; M), dynamic acetylation (+42.011 Da; N-terminus), static TMT6plex (+229.163 Da; any N-terminus), static TMT6plex (+229.163 Da; K), and static carbamidomethyl (+57.021 Da; C). Normalization was turned off. The mass spectrometry proteomics data have been deposited to the ProteomeXchange Consortium via the PRIDE partner repository (*Perez-Riverol et al., 2022*) with the dataset identifier PXD033713 and 10.6019/PXD033713. Protein abundances are reported in *Supplementary file 3*.

Protein abundances were loaded into the R environment (version 4.0.4) and were analyzed using the mineCETSA package (*Dziekan et al., 2020*), which performed normalization, generated log-logistic fits of the temperature profiles, and calculated a Euclidean distance score (*Supplementary file 3*). Individual curve plots were generated using the drc package (*Ritz et al., 2015*) to fit relative abundances at each temperature or concentration.

## Thermal profiling concentration range: Experiment 1

### Parasite harvest and treatment

Concentration range thermal profiling experiments were performed in biological duplicate on different days. Confluent HFF cells in 15 cm dishes were infected with 2–5×10⁷ RH tachyzoites each. When the parasites had fully lysed the host cell monolayer (40–48 hr later), the extracellular parasites passed through a 5 μm filter. The parasite solution was concentrated by centrifugation for 10 min at 1000 x *g*. Parasites were resuspended in 1 ml of wash buffer (5 mM NaCl, 142 mM KCl, 1 mM MgCl₂, 5.6 mM glucose, 25 mM HEPES pH 7.2) and spun again for 10 min at 1000 x *g*.

The parasite pellet was resuspended in 1200 μl of lysis buffer (5 mM NaCl, 142 mM KCl, 1 mM MgCl₂, 5.6 mM glucose, 25 mM HEPES pH 7.2 with 0.8% IGEPAL CA-630, 1 X Halt Protease Inhibitors, and 250 U/ml benzonase) and subjected to three freeze-thaw cycles. An equivalent volume of parasite lysate was combined with 2 X [Ca²⁺]$_{free}$ buffers to attain the final concentrations: 0 nM, 10 nM, 100 nM, 250 nM, 500 nM, 750 nM, 1 μM, 10 μM, 100 μM, and 1 mM. The solutions were aliquoted into two PCR tubes and were incubated for 5 min at 5% CO₂/37 °C. The tubes were placed on heat blocks pre-warmed at 54 °C or 58 °C for 3 min and were then immediately placed on ice. The lysates were transferred to a TLA-100 rotor and were spun at 100,000 x g for 20 min at 4 °C in a Beckman Ultra MAX benchtop ultracentrifuge. The solution, containing the soluble protein fraction, was removed for further processing.

### MS sample preparation and data acquisition

Samples were prepared for mass spectrometry as described in the Temperature Range methods. MS data was acquired using the same instrument and methods as described above. Raw files were searched in Proteome Discoverer 2.4 using the search parameters described in the Temperature Range methods. Normalization was performed in the Proteome Discoverer software. The mass spectrometry proteomics data have been deposited to the ProteomeXchange Consortium via the PRIDE partner repository (*Perez-Riverol et al., 2022*) with the dataset identifier PXD033642 and 10.6019/PXD033642. Protein abundances are reported in *Supplementary file 4*.

## Thermal profiling concentration range: Experiment 2

### Parasite harvest and treatment

Parasites and host cells were grown for one week in SILAC media and were considered separate biological replicates. SILAC media was prepared from DMEM for SILAC (Thermo Fisher 88364), 10% dialyzed FBS (Thermo Fisher A3382001) and L-Arginine HCl/L-Lysine-2-HCl (Thermo Fisher 89989 and 89987) or ¹³C₆¹⁵N₄ L-Arginine HCl/¹³C₆¹⁵N₂ L-Lysine-2HCl (Thermo Fisher 89990 and 88209). The parasites were harvested, lysed, and treated as described in the Concentration range: Experiment 1 procedure. The lysates were incubated on heat blocks pre-warmed at 50 °C, 54 °C, or 58 °C for 3 min and were then immediately placed on ice.

Insoluble aggregates were removed by filtration as described in *Herneisen and Lourido, 2021*. In brief, the lysates were applied to a pre-equilibrated 96-well filter plate (Millipore MSHVN4510) and were spun at 500 x *g* for 5 min. The concentrations of the filtrates containing soluble fractions were quantified with a DC assay. The heavy and light samples were combined at 1:1 wt/wt, yielding an estimated 50 μg total per concentration.

### Protein cleanup and digestion

Proteins were reduced with 5 mM TCEP for 20 min at 50 °C. Alkylation of cysteines was performed with 15 mM MMTS for 10 min at room temperature. The samples were precipitated in 80% ethanol and were washed using the SP3 protocol as described in the Temperature range procedure. After the final wash, the samples were resuspended in 35 μl of digest buffer (50 mM TEAB) and 1 μg Trypsin. Digestion proceeded overnight at 37 °C on a thermomixer shaking at 1000 rpm. The beads were magnetically separated, and the peptide eluatse were removed from further analysis.

### TMT10plex labeling

Peptides were labeled according to the manufacturer's protocol (Thermo Scientific 90111), with the following modifications. The eluates were combined with 100 μg of TMT10plex reagent in 15 μl of

acetonitrile, for an estimated 1:2 w/w peptide:tag labeling reaction. The labeling proceeded for 1 hr at room temperature and was quenched for 15 min with 5% hydroxylamine. The samples were then pooled, flash-frozen, and lyophilized to dryness.

## Fractionation

Samples were fractionated with the Pierce High pH Reversed-Phase Peptide Fractionation Kit according to the manufacturer's instructions for TMT-labeled peptides.

## MS data acquisition

The fractions were lyophilized and resuspended in 10–20 µl of 0.1% formic acid for MS analysis and were analyzed on an Exploris 480 Orbitrap mass spectrometer equipped with a FAIMS Pro source (*Bekker-Jensen et al., 2020*) connected to an EASY-nLC chromatography system as described above. Peptides were separated at 300 nl/min on a gradient of 6–21% B for 41 min, 21–36% B for 20 min, 36–50% B for 10 min, and 50 to 100% B over 15 min. The orbitrap and FAIMS were operated in positive ion mode with a positive ion voltage of 1800 V; with an ion transfer tube temperature of 270 °C; using standard FAIMS resolution and compensation voltages of –50 and –65 V (injection 1) or –40 and –60 (injection 2). Full scan spectra were acquired in profile mode at a resolution of 120,000, with a scan range of 350–1200 m/z, automatically determined maximum fill time, standard AGC target, intensity threshold of $5 \times 10^3$, 2–5 charge state, and dynamic exclusion of 30 s with a cycle time of 2 s between master scans. MS2 spectra were generated with a HCD collision energy of 36 at a resolution of 30,000 using TurboTMT settings with a first mass at 110 m/z, an isolation window of 0.7 m/z, standard AGC target, and auto injection time.

## Thermal profiling concentration range analysis

Raw files were analyzed in Proteome Discoverer 2.4 (Thermo Fisher Scientific) to generate peak lists and protein and peptide IDs using Sequest HT (Thermo Fisher Scientific) and the ToxoDB release49 GT1 protein database. The search included the following post-translational modifications for the first biological replicate, which was collected from parasites grown in light SILAC medium: dynamic phosphorylation (+79.966 Da; S, T, Y), dynamic oxidation (+15.995 Da; M), dynamic acetylation (+42.011 Da; N-terminus), static TMT6plex (+229.163 Da; any N-terminus), static TMT6plex (+229.163 Da; K), and static methylthio (+45.988 Da; C). The heavy samples were searched for the additional modifications Lys8-TMT6plex (+237.177 Da; K) and static Label:$^{13}$C$_6$$^{15}$N$_4$ (+10.008 Da; R). The mass spectrometry proteomics data have been deposited to the ProteomeXchange Consortium via the PRIDE partner repository (*Perez-Riverol et al., 2022*) with the dataset identifier PXD033650 and 10.6019/PXD033650. Protein abundances are reported in *Supplementary file 4*.

Protein abundances from Proteome Discoverer 2.4 were loaded into the R environment (version 4.0.4) and were analyzed using the mineCETSA package (*Dziekan et al., 2020*). The data were not further normalized using the package, as normalization had already been performed by the Proteome Discoverer software. Log-logistic fitting of the relative abundance profiles was performed using default settings. The AUC reflects the mean stability change when proteins were detected in both replicates of an experiment. If proteins were not detected, the AUC was calculated from one replicate. Proteins were considered Ca$^{2+}$-responsive if the curve-fit parameters had an R$^2$ >0.8 and AUC two modified Z scores from the median. The two concentration range experiments were analyzed separately. Individual curve plots were generated using the drc package (*Ritz et al., 2015*) to fit relative abundances at each temperature or concentration.

## Enrichment analysis

Sets of gene ontology terms from the differentially regulated (or Ca$^{2+}$-responsive) and background proteome (all proteins with quantification values in each mass spectrometry experiment) were downloaded from ToxoDB.org (Molecular Function, Computed evidence, P-Value cutoff set to 1). Gene ontology terms were tested for enrichment across all gene ontology terms identified in the background proteome. A p value for the likelihood of a given enrichment to have occurred by chance was obtained using a hypergeometric test.

## Immunoblotting

Samples were prepared as described in the thermal profile concentration range or phosphoproteomics procedures prior to proteomics sample preparation. The samples, which had already been treated with benzonase, were combined with 5 X laemmli sample buffer (10% SDS, 50% glycerol, 300 mM Tris HCl pH 6.8, 0.05% bromophenol blue, 5% beta-mercaptoethanol) and were incubated at 37 °C for 10 min. The samples were then run on precast 4–15% SDS gels (BioRad) and were transferred overnight onto nitrocellulose membranes at 4 °C and 30 mA in 25 mM TrisHCl, 192 mM glycine, and 20% methanol. Blocking and antibody incubations were performed in 5% milk in TBS-T for 1 hr at room temperature. The membrane was washed three times with TBS-T between antibody incubations. Imaging was performed with the LICOR Odyssey CLx.

## Immunofluorescence assays

Confluent HFFs seeded onto coverslips were infected with extracellular parasites and were grown at 37 °C/5% $CO_2$. Approximately 21 hr later, IAA or a vehicle solution of PBS were added to the wells to a final concentration of 500 μM where indicated. At 24 hr post-infection, the media was aspirated, and coverslips were fixed in 4% formaldehyde in PBS. Following three washes in PBS, the fixed cells were permeabilized with 0.25% triton for 10 min at room temperature. Residual permeabilization solution was removed with three washes of PBS. The coverslips were incubated in blocking solution (5% IFS/5% NGS in PBS) for 10 min at room temperature, followed by a 60-min incubation in primary antibody solution. An anti-CDPK1 antibody (Covance) was used as a parasite counterstain (*Waldman et al., 2020*). After three washes with PBS, the coverslips were incubated in blocking solution at room temperature for 5 min, followed by a 30-min incubation in secondary antibody solution. The coverslips were washed three times in PBS and once in water. Coverslips were mounted with Prolong Diamond and were set for 30 min at 37 °C. Imaging was performed with the Nikon Ti Eclipse and NIS Elements software package.

## Invasion assays

Confluent HFFs seeded onto coverslips were incubated with 5× $10^6$ extracellular parasites for 60 min at 37 °C/5% $CO_2$. The coverslips were washed four six times with PBS and were fixed for 10 min at room temperature with 4% formaldehyde in PBS. The coverslips were incubated in blocking solution (1% BSA in PBS) for 10 min. Extracellular parasites were stained with mouse anti-SAG1 for 30 min at room temperature. Following permeabilization with 0.25% triton-X100 for 10 min, the coverslips were incubated with guinea-pig anti-CDPK1 as a parasite counterstain for 30 min at room temperature. The coverslips were incubated with a secondary antibody solution containing Hoechst and were mounted on coverglass with Prolong Diamond. The number of parasites invaded was calculated by normalizing the number of intracellular, invaded parasites to host cell nuclei in a field of view. Five random fields of view were imaged per coverslip. Each experiment was performed in technical duplicate.

## Egress assays

Automated, plate-based egress assays were performed as previously described (*Shortt and Lourido, 2020*). In brief, HFF monolayers in a clear-bottomed 96-well plate were infected with 7.5× $10^4$ or 1× $10^5$ parasites of the TIR1 or PP1-AID strains, respectively. IAA or PBS were added to a final concentration of 500 μM 20 hr later. After 3 hr, the media was exchanged for FluoroBrite supplemented with 3% calf serum. Three images were taken before zaprinast (final concentration 500 μM) or A23187 (final concentration 8 μM) and DAPI (final concentration 5 ng /mL) were injected. Imaging of DAPI-stained host cell nuclei continued for 9 additional minutes before 1% Triton X-100 was injected into all wells to determine the total number of host cell nuclei. Imaging was performed at 37 °C and 5% $CO_2$ using a Biotek Cytation 3. Results are the mean of two wells per condition and are representative of three independent experiments.

## Replication assays

Parasites were inoculated onto coverslips containing HFFs. After 1 hr, the media was aspirated and replaced with media containing 500 μM IAA or PBS vehicle. At 24 hr post-IAA addition, the coverslips with intracellular parasites were fixed, permeabilized, and stained with CDPK1 antibody and Hoechst as described under 'Immunofluorescence assays'. For each sample, multiple fields of view

were acquired with an Eclipse Ti microscope (Nikon) and the number of nuclei per vacuole were calculated from the full field of view (at least 100 vacuoles). Results are the mean of three independent experiments.

## Plaque assays

A total of 500 parasites were inoculated into 12-well plates of HFFs maintained in D10 and allowed to grow undisturbed for 7 days. IAA or vehicle (PBS) was added to a final concentration of 100 µM. Plates were washed with PBS and fixed for 10 min at room temperature with 100% ethanol. Staining was performed for 5 min at room temperature with crystal violet solution, followed by two washes with PBS, one wash with water, and drying.

## Live microscopy

PP1-mNG parasites were grown in HFFs in glass-bottom 35 mm dishes (Ibidi) for 24 hours. The media was decanted and the dish was washed once with 1 ml Ringer's buffer (155 mM NaCl, 2 mM CaCl$_2$, 3 mM KCl, 1 mM MgCl$_2$, 3 mM NaH$_2$PO$_4$, 10 mM HEPES, 10 mM glucose). Parasites were stimulated to egress with 500 µM zaprinast or 4 µM A23187 in Ringer's buffer (155 mM NaCl, 2 mM CaCl$_2$, 3 mM KCl, 1 mM MgCl$_2$, 3 mM NaH$_2$PO$_4$, 10 mM HEPES, 10 mM glucose) supplemented with 1% FBS (v/v) and recorded every 2 s for 300 s using an Eclipse Ti microscope (Nikon) with an enclosure maintained at 37 °C.

## Cytosolic Ca$^{2+}$ measurements with FURA-2AM

Fura-2 AM loading of *T. gondii* tachyzoites was done as described previously (*Moreno and Zhong, 1996*). Briefly, freshly collected parasites (TIR1 or PP1-AID parasites treated with IAA for 5 hr) were washed twice with buffer A plus glucose or BAG (116 mM NaCL, 5.4 mM KCL, 0.8 mM MgSO$_4$.7H$_2$O, 50 mM Hepes pH 7.3, 5 mM Glucose) by centrifugation (706 x g for 10 min) and re-suspended to a final density of 1 x l0$^9$ parasites/ml in loading buffer (BAG plus 1.5% sucrose, and 5 µM Fura-2 AM). The suspension was incubated for 26 min at 26 °C with mild agitation. Subsequently, parasites were washed twice by centrifugation (2000 x g for 2 min) with BAG to remove extracellular dye, re-suspended to a final density of 1x10$^9$ parasites per ml in BAG and kept on ice. Parasites are viable for a few hours under these conditions. For fluorescence measurements, 2x10$^7$ parasites/mL were placed in a cuvette with 2.5 mL of BAG. Fluorescence measurements were done in a Hitachi F-7100 fluorescence spectrometer using the Fura 2 conditions for excitation (340 and 380 nm) and emission (510 nm). The Fura-2 fluorescence response to Ca$^{2+}$ was calibrated from the ratio of 340/380 nm fluorescence values after subtraction of background fluorescence at 340 and 380 nm as described previously (*Grynkiewicz et al., 1985*). Ca$^{2+}$ release rate is the change in Ca$^{2+}$ concentration during the initial 20 s after compound addition. Delta [Ca$^{2+}$] was calculated by the difference between the higher Ca$^{2+}$ peak and basal Ca$^{2+}$.

## Acknowledgements

We thank Emily Shortt for assistance with cell culture, Eric Spooner of the Whitehead Proteomics Core facility for assistance with sample preparation, and Tyler Smith for providing assistance with tagging vector design. We thank L David Sibley for the MIC2 and SAG1 antibodies, Dominique Soldati-Favre for the GAP45 antibody, and Marc-Jan Gubbels for the TUB1 antibody. This research was supported by funds from National Institutes of Health grants to SL (R01AI144369) and SNJM (R01AI128356 and R21AI15493), and a National Science Foundation Graduate Research Fellowship to ALH (174530).

# Additional information

## Competing interests

Sebastian Lourido: Reviewing editor, *eLife*. The other authors declare that no competing interests exist.

## Funding

| Funder | Grant reference number | Author |
|---|---|---|
| National Institutes of Health | R01AI144369 | Sebastian Lourido |
| National Science Foundation | 174530 | Alice L Herneisen |
| National Institutes of Health | R01AI128356 | Silvia NJ Moreno |
| National Institutes of Health | R21AI15493 | Silvia NJ Moreno |

The funders had no role in study design, data collection and interpretation, or the decision to submit the work for publication.

## Author contributions

Alice L Herneisen, Conceptualization, Data curation, Formal analysis, Investigation, Visualization, Writing - original draft, Writing - review and editing; Zhu-Hong Li, Investigation; Alex W Chan, Methodology; Silvia NJ Moreno, Resources, Supervision, Writing - review and editing; Sebastian Lourido, Conceptualization, Resources, Supervision, Funding acquisition, Writing - review and editing

## Author ORCIDs

Alice L Herneisen ⓘ http://orcid.org/0000-0003-3368-0893
Silvia NJ Moreno ⓘ http://orcid.org/0000-0002-2041-6295
Sebastian Lourido ⓘ http://orcid.org/0000-0002-5237-1095

## Decision letter and Author response

Decision letter https://doi.org/10.7554/eLife.80336.sa1
Author response https://doi.org/10.7554/eLife.80336.sa2

# Additional files

## Supplementary files

• Supplementary file 1. Sub-minute phosphoproteomics time course protein and abundance assignments from Proteome Discoverer 2.4.

• Supplementary file 2. Sub-minute phosphoproteomics time course peptide and abundance assignments from Proteome Discoverer 2.4. Mclust cluster assignments (column 118) of phosphopeptides dynamically changing during zaprinast treatment.

• Supplementary file 3. Data pertaining to the temperature range thermal profiling experiment. 1. Protein and abundance assignments from Proteome Discoverer 2.4 for samples with 0 µM Ca$^{2+}$, replicate 1. 2. Protein and abundance assignments from Proteome Discoverer 2.4 for samples with 0 µM Ca$^{2+}$, replicate 2. 3. Protein and abundance assignments from Proteome Discoverer 2.4 for samples with 10 µM Ca$^{2+}$, replicate 1. 4. Protein and abundance assignments from Proteome Discoverer 2.4 for samples with 10 µM Ca$^{2+}$, replicate 2. 5. Curve fit output from the mineCETSA package. 6. Area under the euclidean distance score calculations from the mineCETSA package.

• Supplementary file 4. Data pertaining to the concentration range thermal profiling experiments. 1. Protein and abundance assignments from Proteome Discoverer 2.4 for Experiment 1 samples with 54 °C, replicate 1. 2. Protein and abundance assignments from Proteome Discoverer 2.4 for Experiment 1 samples with 54 °C, replicate 2. 3. Protein and abundance assignments from Proteome Discoverer 2.4 for Experiment 1 samples with 58 °C, replicate 1. 4. Protein and abundance assignments from Proteome Discoverer 2.4 for Experiment 1 samples with 58 °C, replicate 2. 5. Curve fit output for concentration range Experiment 1 from the mineCETSA package. 6. Area under the curve score calculations from the mineCETSA package for concentration range Experiment 2. 7. Protein and abundance assignments from Proteome Discoverer 2.4 for Experiment 2 samples with 50 °C, replicate 1. 8. Protein and abundance assignments from Proteome Discoverer 2.4 for Experiment 2 samples with 50 °C, replicate 2. 8. Protein and abundance assignments from Proteome Discoverer 2.4 for Experiment 2 samples with 54 °C, replicate 1. 9. Protein and abundance assignments from Proteome Discoverer 2.4 for Experiment 2 samples with 54 °C, replicate 2. 10. Protein and abundance assignments from Proteome Discoverer 2.4 for Experiment 2 samples

with 58 °C, replicate 1. 11. Protein and abundance assignments from Proteome Discoverer 2.4 for Experiment 2 samples with 58 °C, replicate 2. 12. Curve fit output for concentration range Experiment 2 from the mineCETSA package. 13. Area under the curve score calculations from the mineCETSA package for concentration range Experiment 2.

• Supplementary file 5. PP1 depletion zaprinast phosphoproteomics time course protein and abundance assignments from Proteome Discoverer 2.4.

• Supplementary file 6. PP1 depletion zaprinast phosphoproteomics time course peptide and abundance assignments from Proteome Discoverer 2.4. Mclust cluster assignments (column 2) of phosphopeptides dynamically changing during zaprinast treatment when PP1 is depleted.

• Supplementary file 7. Sequences and accessions of oligonucleotides and plasmids used in this study.

• MDAR checklist

## Data availability

All mass spectrometry proteomics data have been deposited to the ProteomeXchange Consortium via the PRIDE partner repository with the dataset identifier PXD033765 and https://doi.org/10.6019/PXD033765. All other information is provided in the supplementary files.

The following dataset was generated:

| Author(s) | Year | Dataset title | Dataset URL | Database and Identifier |
|---|---|---|---|---|
| Herneisen AL | 2022 | De novo mapping of the apicomplexan Ca2+-responsive phosphoproteome | https://www.ebi.ac.uk/pride/archive/projects/PXD033765 | PRIDE, PXD033765 |

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
