## [Editor Report]

Herneisen et al., provide a comprehensive and thorough exploration of Ca^2+^ responsive changes in the Toxoplasma proteome and the resulting phosphorylation events during the transition from intracellular residing parasites to egress from the host cell. Furthermore, a novel temperature stability profiling method of all proteins responding to Ca^2+^ concentration with a change in stability is a novel applicable tool that here is used to map Ca^2+^-responsive proteins in the parasites. They provide a compelling analysis of the complex data and carefully validate their findings using genetics and cell biology. This work is of the highest quality in the field.

---

## [Decision Letter]

**Decision letter after peer review:**

Thank you for submitting your article "de novo mapping of the apicomplexan ca^2+^ -responsive proteome" for consideration by *eLife*. Your article has been reviewed by 3 peer reviewers, and the evaluation has been overseen by a Reviewing Editor and Dominique Soldati-Favre as the Senior Editor. The following individual involved in the review of your submission has agreed to reveal their identity: Christopher J Tonkin (Reviewer #1).

Essential revisions:

Overall, the reviewers are very supportive of the work and are praising the superb articulation of the manuscript. Nevertheless, they have identified some weaknesses that deserve your attention. However, following the consultation, the consensus was that given the already significant body of work, we would not require a more deep analysis of the proteomic datasets and more experiments to tackle PP1 regulation/function. Instead, the major concerns could be addressed as follows:

i) Highlight possible links, if any, between the phosphoproteome survey and the ca^2+^ TPP.

ii) A high-level discussion with similar phosphoproteomic surveys in Toxo.

iii) A more careful wording about the regulation/function of PP1.

iv) The title should be less general.

*Reviewer #1 (Recommendations for the authors):*

I only have small comments to make about this paper:

1. Abstract: It is stated in the abstract ' ca^2+^ signaling pathways have been repurposed in these eukaryotic pathogens to regulate parasite-specific cellular processes governing the transition between the replicative and lytic phases of the infectious cycle.'

I would say that ca^2+^ signalling has been implicated in more than this. CDPK7 for example appears to be involved in replication and CDPK2 in the regulation of amylopectin levels. I think it would be better to be a bit broader in the interpretation of what ca^2+^ signaling does (at least as we understand it now).

2. Line 436: The authors state that both subunits of PKA 'subunits' are the only known cAMP receptors in Toxo. Do the authors mean catalytic and regulatory subunits? If so, this is not true. Sugi et al., mBio 2016 identify another PKA orthologue (presumably that has its own regulatory subunit) and show it is somehow involved in negatively regulating bradyzoite development. The authors should expand their statement to include this study.

*Reviewer #2 (Recommendations for the authors):*

Phosphoproteome

I think the authors should be more cautious when mentioning ca^2+^ signalling processes in general, which is at times possibly misleading for example in the main title and the heading of the first result section:

1. As mentioned in the text zaprinast is elevating cGMP levels that, in turn, lead to elevated intracellular ca^2+^ levels. However, there are likely ca^2+^-independent pathways downstream of cGMP that are included in this dataset.

2. On the other hand, there are likely a lot of other ca^2+^-dependent phosphorylation events that are not downstream of cGMP during the parasite development.

The authors do not cite a similar work by Dominicus et al. I think it would be important to cite this work and possibly compare the datasets. I understand this represents significant work but this would also likely highlight important proteins with more confidence.

Thermal proteome profiling

The analysis seems to be biased towards egress and/or invasion proteins without a general unbiased analysis. For example, I was surprised that centrins did not appear in the main text, figure, or table. On the other hand, the C2-domain-containing protein DOC2 involved in egress is not discussed. Is this just due to the authors' interest or are these proteins not identified by the TPP? If this is the case it may be interesting to discuss "missing" proteins.

TPP + phosphoproteome

Although I understand that there is no direct obvious functional link between the zaprinast-dependent phosphoproteome and the ca^2+^ TPP, are there proteins that are highlighted by the two analyses? I am not sure a comparison between the two datasets would make sense but I would suggest highlighting in Table 1 whether the discussed protein is differentially regulated upon zaprinast stimulation or predicted to by to ca^2+^ by the TPP, this could help the reader tremendously.

The authors also use "calcium-responsive" or "calcium-regulated" which can be slightly confusing as this suggests, to me, a functional modification. Wouldn't it be better to use "calcium-binding"?

PP1 analysis

The authors decide to investigate the role of PP1 during egress as it is stabilised by calcium, which could not be predicted and is not reported for other eukaryotic PP1. However, while the authors investigate the role of PP1 in egress, they do not confirm the role of calcium stabilisation in the function of PP1 in egress, which would be important to validate the link between ca^2+^ and PP1. This additionally raises the question as to whether this property is specific to the function of TgPP1 compared with eukaryotic PP1. Could divergences in TgPP1 possibly point to the ca^2+^-binding domain?

On the other hand, the authors propose a role of PP1 in calcium entry. This was based on recent findings that Toxoplasma exhibits ca^2+^ activated ca^2+^ entry upon zaprinast treatment. Previous experiments have shown that zaprinast exposure induces two rapid calcium peaks in intracellular parasites, the second peak is dependent on extracellular calcium. While the general calcium response is clearly delayed upon PP1 depletion I am not sure to see any distinguishable peak. Could the authors explain better their interpretation of the results? In *Plasmodium falciparum* schizonts, PP1 was suggested to regulate basal cGMP levels. I think the authors should investigate this possibility in Toxoplasma, as this could possibly confound the interpretation of the experiments shown in Figure 7A-C. It also seems that basal calcium levels are lower in PP1-AID extracellular tachyzoites, could this affect the ca^2+^ uptake?

Others

I would suggest avoiding generalising the findings to Apicomplexa parasites as experiments are performed in Toxoplasma only.

*Reviewer #3 (Recommendations for the authors):*

Besides the concern regarding the description of the PP1 dynamics mentioned in the Public Review, other points that need attention are the over stylized presentation of data that prevents clear interpretation and communication of the message, whereas some data presented is barely discussed and interpreted in the text (the result of the data avalanche).

Specific points

1. Line 126/7. The role of ERK7 is primarily more geared toward cell division (conoid/cytoskeleton stability), which as a secondary effect impacts motility. Although technically the statement is not untrue, relating ERK7 to cell division would be more appropriate.

2. Some results are discussed without showing the data, for instance, the structural modeling of TGGT1_209950 to identify whether it is related to calsequestrin is not included. This seems to be a tentative association at best, based on how it is currently described and presented. Either more details or more cautious wording to an interpretation/conclusion are recommended.

3. With the apical annuli now assigned to a role in dense granule secretion, the phosphorylation of AAP2 and AAP5 seems to connect the ca^2+^-dependence of dense granule secretion: PMID: 30673152

4. Line 290-294. *T. gondii* GPM1 has been directly studied genetically and in the context of ca^2+^, where, interestingly, the only phenotype was A12387 microneme secretion; PMID: 29202046

5. It is somewhat surprising that many EF-hand proteins do not show changes in thermal stability upon changing [ca^2+^]. Adding a short discussion on potential mechanisms would be useful (e.g. technical issue, protein abundance, or a physiological issue?).

6. Although PI-PLC is mentioned as an example of a protein with a C2 domain, other C2 domain proteins are not systematically evaluated; e.g. a figure in 3D for the EF hands should be considered for the C2 domain proteins.

7. The interpretation of localization shown in Figure 4B is exceptionally brief; "These ca^2+^-responsive proteins localized to diverse structures.." some annotation is provided in the legend (looks like the 'green arrow' structure is more consistent with the residual body than to the basal pole), but a better interpretation of what is shown is warranted in the main text, e.g. the nuclear localizations.

8. The use of the term "holoenzyme" (i.e. association with distinct protein complexes to explain dynamic and pleiotrophic localization) is speculative and derived by inference from other eukaryotes. Although this is a feasible model, the hypothesis is not directly tested here in *T. gondii*. Alternatives are changes in membrane binding capacities e.g. changes in lipid in membranes, or PP1 modification or structural changes are very reasonable alternative models, given the role of lipid/membrane composition changes in triggering egress.

9. There are no legends for the videos.

10. Figure 5J. Egress assay. These look like fitted lines of 'host nuclei DAPI+ counts; please add the primary data points (or otherwise explain what is plotted in more detail).

11. Figure 7B/E and legend. I do not see a transparent line in the figure; it is not explained what the grey, yellow and blue lines represent. How many lines are shown, and how do they represent 3 different biological reps?

12. Figure 7C/F and Figure 7 Supplement. Different shades of grey (let alone transparency) as referred to in the legend cannot be appreciated. Since there are only 2 solid, large dots, it looks like n=2, which also implies these data are not corresponding with panel B/E, where it says n=3. Also, why are some p values in grey (barely legible) and others in black?

13. Font sizes and colors in the figures are not very conducive to reading without zooming in.

14. A Supplementary Table with primers used to make transgenics should be included; gene annotations on ToxoDB change so geneIDs are not reliable to see which reading frame annotation was used, and it also is hard to evaluate the findings in light of ORF models and genome annotations.

---

## [Author Response]

Essential revisions:Overall, the reviewers are very supportive of the work and are praising the superb articulation of the manuscript. Nevertheless, they have identified some weaknesses that deserve your attention. However, following the consultation, the consensus was that given the already significant body of work, we would not require a more deep analysis of the proteomic datasets and more experiments to tackle PP1 regulation/function. Instead, the major concerns could be addressed as follows:i) Highlight possible links, if any, between the phosphoproteome survey and the ca^2+^ TPP

As suggested by Reviewer 2, we have added columns to Table 1 specifying whether proteins were detected in the phosphoproteomic or thermal profiling surveys.

We also highlighted links between the phosphoproteomic and thermal profiling analyses throughout the discussion. Some nuance is required in framing these connections, as the datasets were profiled across different dimensions (thermal stability at protein level vs. kinetics at peptide level). The designation of a peptide as *dynamic* or protein as *calcium-responsive* ultimately derives from the evaluation criteria. We have clearly described the metrics and thresholds we relied on for our analysis, but we do not intend that analysis to be static. We encourage readers to draw their own thresholds or examine abundance profiles on a case-by-case basis. Alternative packages and software may be used for clustering (e.g. PMID: 27348712) and thermal profiling analysis (e.g. PMID: 26379230), and more sophisticated and emergent analysis approaches may provide more insights from these datasets that we had originally envisioned (e.g. doi.org/10.1101/2022.06.10.495491)

ii) A high-level discussion with similar phosphoproteomic surveys in Toxo.

We have expanded the discussion to reference other phosphoproteomic studies that probed enzymes involved in the kinetic phase of the lytic cycle. Comparisons between the phosphoproteomes of parasites treated with phosphodiesterase inhibitor (this study) or calcium ionophore (Dominicus et al., 2021) may reveal commonalities in the pathways stimulated by the tow agonists. For example, Dominicus et al., note high levels of agreement between differentially up-regulated phosphosites upon BIPPO and A23187 treatment.

One of the reviewers specifically mentions the preprint by Dominicus et al., 2021. That study presented multiple phosphoproteomes of *T. gondii* parasites, summarized as (i) a comparison between treatments (PDE inhibitor or calcium ionophore) of different strains at a single time point post-stimulation and (ii) a time-resolved phosphoproteome of two strains in the minute following A23187 treatment. The datasets generated by the Dominicus et al., study are not in the public domain at the time of resubmission, as no supplementary files or PRIDE accessions were provided with the preprint**.** A direct comparison between the two datasets could be informative, perhaps by revealing central nodes in the signaling networks precipitating motility, as well as treatment-specific variance in the topology of these pathways. For example, the body of work we present here suggests that PP1 may phosphorylate key targets during zaprinast-induced but not ionophore-induced egress. The Dominicus et al., report a 91% overlap of up-regulated phosphosites following BIPPO and A23187 treatment at a single time point–and a comparatively poor overlap of 58% of down-regulated sites, which may be the substrates of calcium-activated phosphatases. Metanalyses of the two studies, will be able to resolve several important questions. Could the differential function of PP1 under the two treatment regimes account for the low overlap of down-regulated sites? Do any of the phosphopeptides or phosphoproteins down-regulated by BIPPO but not A23187 treatment overlap with sites identified as PP1-dependent in our dataset?

At a high level, we compare some of the features and conclusions of the two studies here. Both studies generated time-resolved phosphoproteomes within a minute of treatment by signaling agonists used to stimulate *T. gondii* egress. Dominicus et al., treated intracellular parasites with the calcium ionophore A23187, which moves calcium ions across membranes, including host cell membranes. By contrast, we treated extracellular parasites with zaprinast, which initiates cGMP signaling and release from parasite calcium stores. Given the large differences between ion concentrations in the host cell and extracellular environments, the initial signaling states of the parasites were likely divergent between the studies; however, each study normalized dynamics to an internal control. Both studies employed unbiased mixture-model clustering on differentially regulated phosphopeptides. Collectively, the studies identify consistent nodes of crosstalk between second messenger signaling pathways. Although cyclic nucleotide pathways are traditionally thought of as upstream of calcium release in apicomplexans, the Dominicus et al., study pointed toward extensive regulation of enzymes in the cNMP pathways following treatment with calcium ionophore, which likely bypasses some of the regulated steps of calcium release and uptake from parasite stores. PDE1, PDE2, and the guanylyl and adenylyl cyclases were all heavily regulated. We identified some of these enzymes (PDE2, GC, and the CNB protein AAP2) as candidate signaling platforms, as in our study they belonged to several kinetically resolved clusters, suggesting regulation by multiple kinases or phosphatases. Both studies identified regulation of the cAMP pathway by calcium: our thermal profiling approach revealed the calcium-responsive stability change of PKA, and the Dominicus et al., phosphoproteome hints that the adenylyl cyclase may be regulated by CDPK3. Perhaps the calcium-dependent stability of PKA and the calcium-dependent regulation of the adenylyl cyclase are linked. Additional insights will likely emerge upon integration of the studies.

We have summarized these statements in the discussion as follows:

“Advances in proteomics technologies, such as enhanced sensitivity and improved multiplexing methods, are now being leveraged to monitor the sub-minute processes of exflagellation and egress (Dominicus et al., 2021; Invergo et al., 2017). Second- messenger signaling pathways underpin these transitions at distinct developmental stages in diverse apicomplexans, with many kinases and phosphatases functioning at multiple steps in the parasitic life cycle. Meta-analyses of orthogonal proteomic profiles, such as a phosphoproteome tracking ionophore-induced egress (Dominicus et al., 2021), could reveal signatures of signaling subnetworks. A recent study points to extensive regulation of enzymes in the cyclic nucleotide pathways following treatment with calcium ionophore, which likely bypasses some of the regulated steps of calcium release and uptake from parasite stores—PDE1, PDE2, and the guanylyl and adenylyl cyclases were all heavily regulated (Dominicus et al., 2021). We identified some of these enzymes (PDE2, GC, and the CNB protein AAP2) as candidate signaling platforms, as in our study they belonged to several kinetically resolved clusters, suggesting regulation by multiple kinases or phosphatases. “

iii) A more careful wording about the regulation/function of PP1.

We have amended the discussion to more clearly delineate (i) our *working model* for PP1 holoenzyme function(s) during the kinetic phase and (ii) when we make assumptions building on previous reports in the literature. We noted additional experimental work required to assess the model–for example, using genetic pathway analysis to place PP1 more precisely in the cGMP/calcium signaling network–and have emphasized the importance of identifying the phosphatase regulatory subunits for more systematic characterization of the specific holoenzymes involved in the transition the kinetic phase of infection.

iv) The title should be less general.

We have changed the title from the more general de novo *mapping of the apicomplexan ca^2+^-responsive proteome* to “*Temporal and thermal profiling of the* Toxoplasma *proteome implicates parasite Protein Phosphatase 1 in the regulation of ca^2+^-responsive pathways”.*

Beyond the concerns raised by the review team, we have identified and corrected the following errors or omissions in the first submission of the manuscript:

– Figure S7C: we have corrected mislabeling of the x-axis.

– Figure 3D and 3E: we have changed the color of zero counts to gray so that they are easier for the reader to distinguish.

– Figure 7H: We moved the previous panel to the supplement and added in its place summaries of resting cytosolic calcium concentrations and calcium entry rates at the highest concentration tested. (See response to Reviewer 2.)

– We have condensed the number of supplementary files by combining the contents of files 2 and 3 and combining files 7 and 8. We have re-numbered references to table numbers in the text accordingly.

Reviewer #1 (Recommendations for the authors):I only have small comments to make about this paper:1. Abstract: It is stated in the abstract ‘ ca^2+^ signaling pathways have been repurposed in these eukaryotic pathogens to regulate parasite-specific cellular processes governing the transition between the replicative and lytic phases of the infectious cycle.’I would say that ca^2+^ signalling has been implicated in more than this. CDPK7 for example appears to be involved in replication and CDPK2 in the regulation of amylopectin levels. I think it would be better to be a bit broader in the interpretation of what ca^2+^ signaling does (at least as we understand it now).

This is true; relatively little attention has been paid to the role of calcium signal sensors and transducers during the replicative phase of infection. We have now included mentions to CDPK2 and CDPK7 in the introduction, as suggested by the reviewer. As Reviewer 2 also pointed out, the manuscript does focus on proteins that may play a role in rapid transitions. We hope that this resource will inform hypotheses about the role of candidate proteins during cell division and replication. We have broadened the role of calcium signaling in the text to include the replicative phase of the infection as well.

2. Line 436: The authors state that both subunits of PKA 'subunits' are the only known cAMP receptors in Toxo. Do the authors mean catalytic and regulatory subunits? If so, this is not true. Sugi et al., mBio 2016 identify another PKA orthologue (presumably that has its own regulatory subunit) and show it is somehow involved in negatively regulating bradyzoite development. The authors should expand their statement to include this study.

Here “subunits” refers to the catalytic and regulatory subunits of PKA. Biochemical and genetic studies support the model of a cAMP-dependent interaction between PKA-C1 and PKA-R (PMID: 30208022 and 29030485; also cited in text). As referenced here, an additional PKA catalytic subunit, PKA-C3, has been reported to prevent conversion to the bradyzoite stages in cell culture. However, no regulatory subunit of PKA-C3 has been reported. As the PKA regulatory subunit, and not the catalytic subunit, possesses cyclic nucleotide binding domains, it is not evident how PKA-C3 would be regulated by cAMP in *T. gondii*, if at all.

Nonetheless, we have better qualified this statement in the text, instead designating PKA subunits as the “best-characterized” cAMP receptors in *T. gondii*.

Reviewer #2 (Recommendations for the authors):PhosphoproteomeI think the authors should be more cautious when mentioning ca^2+^ signalling processes in general, which is at times possibly misleading for example in the main title and the heading of the first result section:1. As mentioned in the text zaprinast is elevating cGMP levels that, in turn, lead to elevated intracellular ca^2+^ levels. However, there are likely ca^2+^-independent pathways downstream of cGMP that are included in this dataset.

We have now included mention in the results that:

“PKG likely performs functions that extend beyond regulating ca^2+^ stores, for example by mobilizing diacylglycerol and phosphatidic acid (Lourido et al., 2012; Brown et al., 2017; Bullen et al., 2016; Bisio et al., 2019); however, the use or phosphodiesterase inhibitors like zaprinast allows us to stimulate endogenous ca^2+^ release without flooding the cell with ca^2+^, as is the case with ionophores.” to make readers aware of this fact. We also mention in the discussion that “Through the use of zaprinast we stimulate the endogenous regulation of ca^2+^ stores by PKG. Our phosphoproteomic analysis therefore includes ca^2+^-responsive pathways, as well as any other pathways downstream of cGMP or PKG, which may include ca^2+^-independent processes.”

2. On the other hand, there are likely a lot of other ca^2+^-dependent phosphorylation events that are not downstream of cGMP during the parasite development.

Stimulation of the PKG pathway through phosphodiesterase inhibitors remains the best understood method to trigger release of intracellular ca^2+^ stores in a physiologically relevant manner. The cytosolic ca^2+^ surge achieved by zaprinast would be expected to stimulate any ca^2+^-dependent kinase that responds to physiologically relevant levels of ca^2+^. Nevertheless, our discussion of the phosphoproteomic data does not claim to encompass all ca^2+^-dependent phosphorylation events, taking care to refer to those that are downstream of zaprinast, which, even if incomplete, is clearly relevant for the critical process parasite motility.

The authors do not cite a similar work by Dominicus et al. I think it would be important to cite this work and possibly compare the datasets. I understand this represents significant work but this would also likely highlight important proteins with more confidence.

We have cited this preprint. See Essential Revisions, point (ii).

Thermal proteome profilingThe analysis seems to be biased towards egress and/or invasion proteins without a general unbiased analysis. For example, I was surprised that centrins did not appear in the main text, figure, or table. On the other hand, the C2-domain-containing protein DOC2 involved in egress is not discussed. Is this just due to the authors' interest or are these proteins not identified by the TPP? If this is the case it may be interesting to discuss "missing" proteins.

Our presentation of the data had the goal of demonstrating the power of the resources while also maintaining a coherent narrative. Investigators studying calcium-regulated processes in other steps of the lytic cycle have access to the data and may search for proteins of interest on a case-by-case basis. Our ability to quantify proteins with confidence is limited by the sensitivity of LC-MS: low copy-number proteins, proteins difficult to solubilize, and proteins lacking peptides that “fly”, will appear stochastically and without reliable quantification values. Thermal challenge further reduces coverage of the proteome. DOC2.1 is presumed a low copy-number protein and was not abundant enough to yield thermal profiles. The centrins likely reside in structures with low solubility and thus may fail to exhibit sigmoidal melting curves, similarly to CAM1 and CAM2 mentioned in the text. We did detect centrins, although none exhibited thermal profiles reproducible enough to pass our thresholds. CEN1 (TGGT1_247230) trended towards destabilization; CEN2 (TGGT1_250340) trended towards stabilization; CEN3 (TGGT1_260670) trended towards stabilization. The complete datasets are also provided as Supplementary Files for readers to curate based on their own interests.

TPP + phosphoproteomeAlthough I understand that there is no direct obvious functional link between the zaprinast-dependent phosphoproteome and the ca^2+^ TPP, are there proteins that are highlighted by the two analyses? I am not sure a comparison between the two datasets would make sense but I would suggest highlighting in Table 1 whether the discussed protein is differentially regulated upon zaprinast stimulation or predicted to by to ca^2+^ by the TPP, this could help the reader tremendously.

As suggested by Reviewer 2, we have added columns to Table 1 specifying whether proteins were detected in the phosphoproteomic or thermal profiling surveys.

The authors also use "calcium-responsive" or "calcium-regulated" which can be slightly confusing as this suggests, to me, a functional modification. Wouldn't it be better to use "calcium-binding"?

We intentionally avoided the term “calcium-binding” in this case, as we hypothesize that many changes in thermal stability arise indirectly from interactions with calcium, e.g. due to changes in interaction partners, localization or membrane association, and calcium-dependent post-translational modifications (e.g. as mediated by kinases and phosphatases with intrinsic calcium-sensing domains, such as the CDPKs or calcineurin).

PP1 analysisThe authors decide to investigate the role of PP1 during egress as it is stabilised by calcium, which could not be predicted and is not reported for other eukaryotic PP1. However, while the authors investigate the role of PP1 in egress, they do not confirm the role of calcium stabilisation in the function of PP1 in egress, which would be important to validate the link between ca^2+^ and PP1. This additionally raises the question as to whether this property is specific to the function of TgPP1 compared with eukaryotic PP1. Could divergences in TgPP1 possibly point to the ca^2+^-binding domain?

We have grappled with this question for some time: why is PP1 calcium-responsive in *T. gondii*? Our preliminary investigations may be the topic of future work. Rigorously deconvolving PP1 calcium-responsiveness would require recombinant expression and purification, which is beyond the scope of the present manuscript. Instead, we leveraged our global datasets to place PP1 in established apicomplexan signaling pathways.

On the other hand, the authors propose a role of PP1 in calcium entry. This was based on recent findings that Toxoplasma exhibits ca^2+^ activated ca^2+^ entry upon zaprinast treatment. Previous experiments have shown that zaprinast exposure induces two rapid calcium peaks in intracellular parasites, the second peak is dependent on extracellular calcium. While the general calcium response is clearly delayed upon PP1 depletion I am not sure to see any distinguishable peak. Could the authors explain better their interpretation of the results? In *Plasmodium falciparum* schizonts, PP1 was suggested to regulate basal cGMP levels. I think the authors should investigate this possibility in Toxoplasma, as this could possibly confound the interpretation of the experiments shown in Figure 7A-C. It also seems that basal calcium levels are lower in PP1-AID extracellular tachyzoites, could this affect the ca^2+^ uptake?

The two peaks are not visible at a population level, as the responses were not synchronized and there is averaging. Vella et al., 2021 note that the double peaks are visible when tracking individual vacuoles (PMID: 33524795, as cited in text). Because this observation is established, we did not show it here.

We have quantified basal calcium rates in TIR1 and PP1-AID parasites treated with IAA. TIR1 parasites had resting cytosolic calcium concentrations close to the reported range of 70-100 nM for *T. gondii* (PMID: 8573106), whereas PP1-AID parasites had significantly lower basal calcium levels which, as suggested, may contribute to the lower rate of calcium entry in PP1-AID parasites. We have added these data to Figure 7.

We have begun to investigate the relationship between cGMP- and PP1-regulated signaling, but we think these results go beyond the scope of the current work. We have added the following statement in the Discussion:

“Additional work probing the cGMP pathway—for example by directly monitoring cyclic nucleotide levels or leveraging mutants expressing conditional depletion alleles of phosphodiesterases, cyclases, and kinases—is required to precisely place PP1 in the network that primes parasites for the kinetic phase of the infection cycle.”

OthersI would suggest avoiding generalising the findings to Apicomplexa parasites as experiments are performed in Toxoplasma only.

We changed the title to telegraph our analysis of the *T. gondii* proteome. However, our findings on PP1 are congruent with recent studies of the enzyme in the related apicomplexan parasite *Plasmodium* spp. (cited in the text), so we hope our work will inspire other comparative efforts to integrate findings across apicomplexan species.

Reviewer #3 (Recommendations for the authors):Besides the concern regarding the description of the PP1 dynamics mentioned in the Public Review, other points that need attention are the over stylized presentation of data that prevents clear interpretation and communication of the message, whereas some data presented is barely discussed and interpreted in the text (the result of the data avalanche).Specific points1. Line 126/7. The role of ERK7 is primarily more geared toward cell division (conoid/cytoskeleton stability), which as a secondary effect impacts motility. Although technically the statement is not untrue, relating ERK7 to cell division would be more appropriate.

We have updated the text to reflect this comment, stating that “ERK7 regulates conoid and cytoskeletal stability during cell division, with secondary functions in parasite egress, motility, and invasion.”

2. Some results are discussed without showing the data, for instance, the structural modeling of TGGT1_209950 to identify whether it is related to calsequestrin is not included. This seems to be a tentative association at best, based on how it is currently described and presented. Either more details or more cautious wording to an interpretation/conclusion are recommended.

We have added more cautious wording to this part of the text.

3. With the apical annuli now assigned to a role in dense granule secretion, the phosphorylation of AAP2 and AAP5 seems to connect the ca^2+^-dependence of dense granule secretion: PMID: 30673152

This is an interesting working hypothesis. To our knowledge, the relationship between the apical annuli and dense granule secretion has not yet been published or appeared on preprint servers, so we hesitate to speculate without a reference.

4. Line 290-294. *T. gondii* GPM1 has been directly studied genetically and in the context of ca^2+^, where, interestingly, the only phenotype was A12387 microneme secretion; PMID: 29202046

Thank you; we have added this reference to the text.

5. It is somewhat surprising that many EF-hand proteins do not show changes in thermal stability upon changing [ca^2+^]. Adding a short discussion on potential mechanisms would be useful (e.g. technical issue, protein abundance, or a physiological issue?).

This is indeed a limitation of thermal profiling: not all changes in protein state result in detectable stability changes. We have added a short discussion at the end of this section and have referenced a review.

6. Although PI-PLC is mentioned as an example of a protein with a C2 domain, other C2 domain proteins are not systematically evaluated; e.g. a figure in 3D for the EF hands should be considered for the C2 domain proteins.

Only a subset of C2 domains bind calcium, e.g. the C2 domain of PKC. Because PKC is presented as the prototypical C2 domain-containing protein, an erroneous assumption that all C2 domains bind calcium pervades the field. The set of proteins with C2 domains identified in our thermal profiling datasets was too small to make systematic evaluations. We included PI-PLC in our discussion because it also contains an EF hand domain.

7. The interpretation of localization shown in Figure 4B is exceptionally brief; "These ca^2+^-responsive proteins localized to diverse structures.." some annotation is provided in the legend (looks like the 'green arrow' structure is more consistent with the residual body than to the basal pole), but a better interpretation of what is shown is warranted in the main text, e.g. the nuclear localizations.

We have expanded our discussion of the interpretation of localizations of candidates. The candidates were chosen for validation of calcium-responsive stability, rather than characterization. Therefore, we have not performed colocalization studies or biochemical fractionation to definitively assign localizations.

We have substituted reference to the basal pole with residual body, in agreement with the reviewer’s observation.

8. The use of the term "holoenzyme" (i.e. association with distinct protein complexes to explain dynamic and pleiotrophic localization) is speculative and derived by inference from other eukaryotes. Although this is a feasible model, the hypothesis is not directly tested here in *T. gondii*. Alternatives are changes in membrane binding capacities e.g. changes in lipid in membranes, or PP1 modification or structural changes are very reasonable alternative models, given the role of lipid/membrane composition changes in triggering egress.

PP1 holoenzymes have not yet been identified in *T. gondii*. However, holoenzymes have been characterized in related *Plasmodium* species (see PMID: 33036936 and references therein). In *T. gondii*, PP1 was shown to interact with the leucine-rich repeat protein LRR1 (PMID: 17660360), and additional regulatory subunits are likely to exist.

We have added a reference to a review of PP1 holoenzymes in *Plasmodium* (PMID: 33036936) when the concept of holoenzymes is introduced in the text. To acknowledge the formal possibility raised here, in the discussion we have added “Alternatively, PP1 itself may change membrane association or structure.”

9. There are no legends for the videos.

We have added legends for the videos.

10. Figure 5J. Egress assay. These look like fitted lines of 'host nuclei DAPI+ counts; please add the primary data points (or otherwise explain what is plotted in more detail).

Correct. As stated in the figure legend, “Egress was monitored by the number of host cell nuclei stained with DAPI over time and was normalized to egress in the vehicle-treated strain.” A more detailed description of the assay is provided in the Materials and methods section, citing the source (Shortt and Lourido, 2020).

11. Figure 7B/E and legend. I do not see a transparent line in the figure; it is not explained what the grey, yellow and blue lines represent. How many lines are shown, and how do they represent 3 different biological reps?

The opaque/transparent lines in Figure 7 panels B/E correspond to the fluorescence of individual vacuoles over time. Line color corresponds to treatment condition. For clarity, we did not use different colors for biological replicates (this information is encoded in panels 7C/F). The thicker, solid lines correspond to the mean fluorescence of all vacuoles of belonging to the same treatment condition at each time. We have attempted to clearly describe this information in the figure legend.

12. Figure 7C/F and Figure 7 Supplement. Different shades of grey (let alone transparency) as referred to in the legend cannot be appreciated. Since there are only 2 solid, large dots, it looks like n=2, which also implies these data are not corresponding with panel B/E, where it says n=3. Also, why are some p values in grey (barely legible) and others in black?

We have adjusted the figures so that the three shades of gray are more apparent. The medians shown in Figure 7C/F and the supplement correspond to panels B/E, n = 3. In some cases, the medians of the different biological replicates were so close as to appear overlapping; we have added additional jitter to the points to improve visibility. References to transparency are not applicable and came from a previous version of the figure; we have removed this reference from the figure legend. We have changed font colors from gray to black.

13. Font sizes and colors in the figures are not very conducive to reading without zooming in.

We will adhere to the figure preparation guidelines to ensure font sizes are legible.. We have changed the font color from gray to black where possible.

14. A Supplementary Table with primers used to make transgenics should be included; gene annotations on ToxoDB change so geneIDs are not reliable to see which reading frame annotation was used, and it also is hard to evaluate the findings in light of ORF models and genome annotations.

We have added DNA sequences to the Key Resources Table and Supplementary File 7.